# Quantitative Comparison of Power Production and Power Quality Onshore and Offshore: A Case Study from the Eastern U.S.

Rebecca Foody[1], Jacob Coburn[1], Jeanie A. Aird[2], Rebecca J. Barthelmie[2] and Sara C. Pryor[1]

[1]Department of Earth and Atmospheric Science, Cornell University, Ithaca, 14850, United States
[2]Sibley School of Mechanical and Aerospace Engineering, Cornell University, Ithaca, 14850, United States

*Correspondence to*: Sara C. Pryor (sp2279@cornell.edu)

**Abstract.** A major issue in quantifying potential power generation from prospective wind energy sites is the lack of observations from heights relevant to modern wind turbines, particularly for offshore where blade tip heights are projected to increase beyond 250 m. We present analyses of uniquely detailed datasets from LiDAR (Light Detection And Ranging) deployments in New York state and on two buoys in the adjacent New York bight to examine the relative power generation potential and power quality at these on- and off- shore locations. Time series of 10-minute wind power production are computed from these wind speeds using the power curve from the International Energy Agency 15 MW reference wind turbine. Given the relatively close proximity of these LiDAR deployments, they share a common synoptic scale meteorology and seasonal variability with lowest wind speeds in July and August. Time series of power production from the on- and off- shore location are highly spatially correlated with the Spearman rank correlation coefficient dropping below 0.4 for separation distances of approximately 350 km. Hence careful planning of on- and off- shore wind farms (i.e. separation of major plants by > 350 km) can be used reduce the system-wide probability of low wind energy power production. Energy density at 150 m height at the offshore buoys is more than 40% higher and the Weibull scale parameter is 2 ms$^{-1}$ higher than at all but one of the land sites. Analyses of power production time series indicate Annual Energy Production is almost twice as high for the two offshore locations. Further, electrical power production quality is higher from the offshore sites that exhibit a lower amplitude of diurnal variability, plus a lower probability of wind speeds below the cut-in and of ramp events of any magnitude. Despite this and the higher resource, the estimated Levelized Cost of Energy (LCoE) is higher from the offshore sites mainly due to the higher infrastructure costs. Nonetheless, the projected LCoE is highly competitive from all sites considered.

## 1    Introduction

The United States government has set a goal of reaching carbon net neutral emissions from the power generation sector by 2035 and a carbon net neutral economy by 2050 (U.S. White House, 2023). As part of this plan, the U.S. Department of Interior is committed to deploying 30 GW of offshore wind power by 2030 (U.S. Department of the Interior, 2021). However, in 2021, 93% of electrical power produced by global wind turbines was derived from those deployed in onshore rather than offshore wind farms, partly due to the higher investment required for offshore wind power installation (IEA, 2022). Within the U.S., as of the end of 2022, there was over 145 GW of wind energy installed capacity onshore and only 42 MW offshore (American Clean Power, 2023).

Enhanced deployment of wind turbines offshore offers great promise in terms of enhanced renewable energy penetration into the electricity generation portfolio for three primary reasons:

- First, wind speeds tend to be higher and more consistent offshore due to both the lower surface roughness and lack of obstacles and topographic features that extract momentum and reduce both the wind speed and wind resource (Pryor and Barthelmie, 2002). Accordingly, Capacity Factors (CF), which are the ratio of actual annual power generation divided by the theoretical maximum power generation, are typically higher offshore. For example, data

from operating wind farms in Denmark indicate CF from four offshore wind farms with installed capacity (IC) of 160 to 400 MW of 41-53% while CF from smaller onshore wind farms (IC: 16-70 MW) have CF of 28-41% (Enevoldsen and Jacobson, 2021). Within the U.S., the mean CF for onshore wind farms built between 2014 and 2019 is approximately 41% (Wiser et al., 2021). Simulations using numerical models for offshore wind energy lease areas along the U.S. east coast indicate CF above 46% largely as a result of the higher wind speeds offshore (Pryor et al., 2021;Barthelmie et al., 2023).

• Second, there are generally fewer social barriers than exist on land (e.g., competition for land, noise concerns, visual blight, etc.) (Diógenes et al., 2020), although there are considerations regarding co-use (e.g. for commercial fishing and marine navigation) (Stone et al., 2017;Kirkegaard et al., 2023). Further, the onshore resource in available areas may not be sufficient to meet projected needs (Esteban et al., 2011). In this context it is worth noting that the U.S. technical offshore wind capacity exceeds 2000 GW with the potential to produce over 7200 TWh per year, nearly 50 twice current U.S. electricity use (4240 TWh) (Musial et al., 2016).

• Third, many major metropolitan areas are located near coastlines, making offshore wind a convenient energy source (Pryor et al., 2021). The cost of transmission and electricity loss during transmission across high-voltage lines both increase with transportation distance (Bamigbola et al., 2014).

Here we focus on the first of these reasons, and specifically seek to quantify the potential benefit of offshore wind turbine 55 deployments using analyses of uniquely detailed wind profiles from an onshore LiDAR (Light Detection And Ranging) network and an offshore LiDAR network. We use these data sets to quantify and compare three aspects of the wind power generation potential on- and offshore:

1 Wind resource and power production. We present Weibull probability distribution parameters and derive energy density from the wind speed time series and compare and contrast the inferred wind resource at the onshore and 60 offshore sites. We further compute and compare the Annual Energy Production (AEP) from the time series of wind speeds at each LiDAR site using a common wind turbine power curve.

2 Power quality. Intermittency is frequently cited as a barrier to increased wind power integration into the electrical grid (Bistline and Blanford, 2021). We quantify and compare the frequency of zero power production and intensity and probability of so-called ramp events (i.e., rapid changes in wind speed and power production) (DeMarco and 65 Basu, 2018;Pichault et al., 2021) from each onshore and offshore site where LiDARs have been deployed.

3 Predictability and persistence of wind speeds and power production (Haghi et al., 2013;Haslett and Raftery, 1989). Within liberalized electricity markets, wind farm owner/operators bid in advance (e.g. 24 hours in advance) and are charged penalties for any imbalance between the bid and actual production (Pinson et al., 2007). Hence, accurate forecasts of wind generation are important to reduce penalties and maximize revenue (Barthelmie et al., 2008). 70 Persistence models where the power production at some future time is modeled as a function of power production in the recent past is often used as a benchmark forecast against which more sophisticated short-term power production models are compared (Kariniotakis et al., 2004). Also many statistical short-term forecast models are predicated in part on persistence (Zeng and Qiao, 2011) and thus are most skillful when the power production time series exhibits high temporal autocorrelation. We quantify the temporal autocorrelation of power production from each onshore and 75 offshore site and compare the degree to which electrical power production from the onshore and offshore locations differ with respect to persistence and short-term predictability.

We further use these LiDAR measurements to quantify and compare a key driver of wind turbine loading at the on- and off-shore locations:

4 Extreme or anomalous wind shear across the rotor plane. Low-Level Jets (LLJ) are confined wind speed maxima 80 within the lower atmospheric boundary layer (Stensrud, 1996) and are associated with enhanced vertical wind speed (and sometimes directional) shear relative to typical near-logarithmic profiles. LLJ within the wind turbine rotor plane

are associated with higher aerodynamic and structural loading (Gutierrez et al., 2019;Gadde et al., 2021). Analyses of simulations with the Weather Research and Forecasting (WRF) model suggest that offshore coastal regions of the U.S. mid-Atlantic (including the locations of the buoys from which data are presented) generally exhibit a weakly sheared profile across the rotor plane and a relatively low frequency of LLJ (Aird et al., 2022;McCabe and Freedman, 2023). Previous research found LLJ in the lowest 500 m of the atmosphere are most frequent south of Massachusetts and during the summer (8% of all hours) (Aird et al., 2022). They frequently occur at heights that intersect the wind turbine rotor plane, and at wind speeds within typical wind turbine operating ranges. Further, LLJ diagnosed from the WRF output were most intense and have lowest elevation under strong horizontal temperature gradients and lower planetary boundary layer heights. For comparative purposes, data from the NYSM LiDARs are used here to evaluate wind shear across the rotor plane and the occurrence, intensity, and height of LLJ at the onshore locations.

We also analyze the LiDAR data to quantify two other properties of relevance to wind energy integration into the electricity generation supply:

5    Co-variation of wind speeds and power production with varying distance separation (Pryor et al., 2014;Solbrekke et al., 2020). The electric power transmission network in the contiguous U.S. comprises three main interconnections (eastern, western, and Electric Reliability Council of Texas (ERCOT)) and 66 'balancing authorities' that oversee regional operation of the electric grid and are referred to as Regional Transmission Operators (RTOs) or Independent System Operators (ISOs). New York (NY) state currently operates as a single state ISO. NY is both a net importer of electricity and the third most efficient state in terms of energy use per U.S. dollar of economic activity (https://www.eia.gov/state/analysis.php?sid=NY). Careful planning of wind farm locations on and offshore could ensure stable supply of wind-generated electricity into the grid and thus aid the transition from electricity imports and a current dependence on nuclear and natural gas (Eryilmaz et al., 2020). Here we quantify the spatial autocorrelation of power production from each onshore and offshore site where the LiDARs have been deployed to evaluate the decorrelation distance and hence provide guidance regarding optimal spatial scale of wind farm separation (on- and off-shore) for stability of wind power supply.

6    Seasonality and diurnal variability of wind power production (WPP) on- and off-shore for demand matching. Electricity demand varies with the level of economic activity and seasonal heating/cooling requirements which are a function of the regional climate (Castillo et al., 2022;Staffell and Pfenninger, 2018). Generally, electricity demand in the U.S. is minimized between approximately 0400 and 0600 local time (LT), is high between 0800 and 1600 LT, and peaks between 1800 and 2100 LT (Burleyson et al., 2021). Diurnal variability of wind power generation is a function of location and land use but, for example, in ERCOT is highest at night (Kiviluoma et al., 2016), consistent with the expectation based on day-time variations in atmospheric stability caused by changes in net radiation and the surface energy balance. Because the oceans have higher specific heat capacity than land, this scale of variability is typically not present in the far offshore (> 20 km from the coast) (Barthelmie et al., 1996). At the seasonal scale, wind resources and power production in the midlatitudes and specifically the contiguous U.S. tend to peak in between October and April and are lowest in July or August due to pronounced shifts in the storm track and the frequency and intensity of mid-latitude cyclones (Pryor et al., 2020b). Recent research suggests WPP is highest in southeastern Canada and the northeastern U.S. during January and February (Coburn and Pryor, 2023). Thus, finally, we quantify whether electrical power from wind turbines deployed offshore exhibit higher or lower temporal matching with electricity demand in New York state at both the diurnal and seasonal scales.

## 2    Data sources

Here we analyze long term (multi-year) measurements from two major LiDAR (Light Detection And Ranging) field

deployments: the New York State Mesonet (NYSM) onshore LiDAR network and the New York State Energy Research and Development Authority (NYSERDA) floating LiDAR campaign. All of the locations considered here lie within a separation distance of a few hundred kilometers and hence are within the so-called 'macro-beta' scale that is influenced by synoptic scale transitory mid-latitude cyclones (i.e., 400-4,000 km) (Stull, 2017). Thus, the expectation is that all sites will experience a relatively similar synoptic scale meteorological regime and that differences in wind resources, power quality and so forth can be largely attributed to differences in the surface; land versus ocean.

## 2.1 NYSERDA LiDAR buoys

To support development of offshore wind energy, NYSERDA undertook a campaign to deploy LiDAR on buoys near prospective offshore wind lease areas (Optis et al., 2021). Here we present data from two of those locations (Figure 1): the Hudson North E05 buoy is located within the Ocean Winds East (OCS-A 0537) lease area, and the Hudson South E06 buoy is located along the Bight Wind Holdings (OCS-A 0539) lease area (BOEM, 2023). The LiDARs deployed on these buoys are ZephIR ZX300M units. They report mean wind speeds, wind direction, and other properties in 10-minute intervals every 20 m up to a maximum height of 200 m. The performance of different series of these robust LiDARs have been extensively evaluated (Barthelmie et al., 2016;Kelberlau and Mann, 2022;Smith et al., 2006) and best practice has been developed for deployment of LiDARs on floating platforms (Bischoff et al., 2017). The LiDAR from the Hudson North E05 buoy has data available from August 2019 through February 2022. The LiDAR on the Hudson South E06 buoy operated from September 2019 through February 2022 but there is lower data availability during August through November (which is partly due to a temporary break in data collection for repairs to the buoy). The LiDARs have an overall data recovery rate of wind speeds at approximately 140 m above sea level of about 77% for the Hudson North E05 buoy and 67% for the Hudson South E06 buoy.

## 2.2 New York State Mesonet

New York state has also invested in a Mesonet (NYSM) to aid hazard mitigation and disaster preparedness. The NYSM includes a network of 17 profiler stations (Shrestha et al., 2022;Brotzge et al., 2020) (Figure 1). The LiDARs deployed as part of the NYSM are the Leosphere WindCube WLS-100 series Doppler LiDAR (Bingöl et al., 2010;Kumer et al., 2016). These pulsed LiDARs have a vertical range of many kilometers and are also configured to report wind speed and direction measurements every 25 m in 10-minute intervals. The period for which data are available varies by location but is generally from January 2019 to December 2022. The sites listed in alphabetical order with their respective abbreviation and in terms of data availability for wind speeds at 150 m are: Albany (ALBA, 36.7%), Belleville (BELL, 29.8%), Bronx (BRON, 64%), Buffalo (BUFF, 44.0%), Chazy (CHAZ, 53.3%), Clymer (CLYM, 48.4%), East Hampton (EHAM, 68%), Jordan (JORD, 51.6%), Owego (OWEG, 56%), Queens (QUEE, 73%), Red Hook (REDH, 54.8%), Staten Island (STAT, 57%), Stonybrook (STON, 59%), Suffern (SUFF, 36.2%), Tupper Lake (TUPP, 53.6%), Wantagh (WANT, 62%), and Webster (WEBS, 49.3%). For much of the following analyses, only the seven sites with data recovery rates (i.e., wind speeds available at 150 m height) > 55% are included.

A critical determinant of LiDAR-derived wind speed and direction climates is the carrier-to-noise ratio (CNR) used in quality control proceedures. CNR is the ratio of the received carrier strength to the intensity of the received noise. Larger values imply higher measurement accuracy but there is ambiguity in terms of the optimal CNR threshold to ensure high wind climate fidelity. Early research with coherent continuous-wave wind LiDAR proposed use of a –22 dB CNR threshold to screen out periods with unacceptably high wind speed uncertainty (Frehlich, 1996), and this threshold has subsequently been widely adopted (Bischoff et al., 2017). Detailed analyses of measurements to 600 m height with Leosphere WLS70 pulsed Doppler LiDAR relative to sonic anemometers, found use of a –22 dB CNR threshold caused a 7 to 12 % overestimation in the long-term mean wind speed, with the higher discrepancy over coastal and marine sites (Gryning et al., 2016). A more recent study, using data from the Leosphere WLS70 deployed on the FINO platform in the North Sea, found a high sensitivity of the wind rose and

mean wind speed to use of thresholds lower than −29 dB (Gryning and Floors, 2019). That analysis further found that for
heights of 100 to 200 m, application of a −22 dB CNR threshold caused a 12% overestimation of mean wind speed, which
decreased to 9% when a CNR threshold value of –35 dB was applied (Gryning and Floors, 2019). Optimal CNR thresholds
may vary with site conditions and instrument. Use of different thresholds will influence only data quality but also data
availability.

As indicated by the above, all LiDAR data time series are incomplete. The NYSM data sets are particularly biased toward data
availability in the summer months. Thus, in the following additional analyses are performed for the 'best available year',
defined as the 365-day period that has highest data availability computed across both NYSERDA buoys and the seven NYSM
sites. This 'best year' extends from September 19, 2019, at 22:50:00 EST to September 18, 2020, at 22:50:00 EST. Data
availability in each of the nine sites for this period is: BRON (67.0%), EHAM (72.9%), OWEG (69.4%), QUEE (78.7%),
STAT (60.0%), STON (70.1%), WANT (79.3%), Hudson North E05 (92.1%), and Hudson South E06 (86.6%).

Enquiries with the NYSM network operator did not resolve any common root cause for the low data availability from these
LiDARs. Documentation associated with the data set notes the causes as; 'calibration errors; power failures; and/or
communication failures.' And further indicates 'Only manufacturer-developed QA/QC procedures are applied to the data and
there might still be some undetected errors.' (readme accessible from
http://www.nysmesonet.org/networks/profiler#stid=prof_alba).

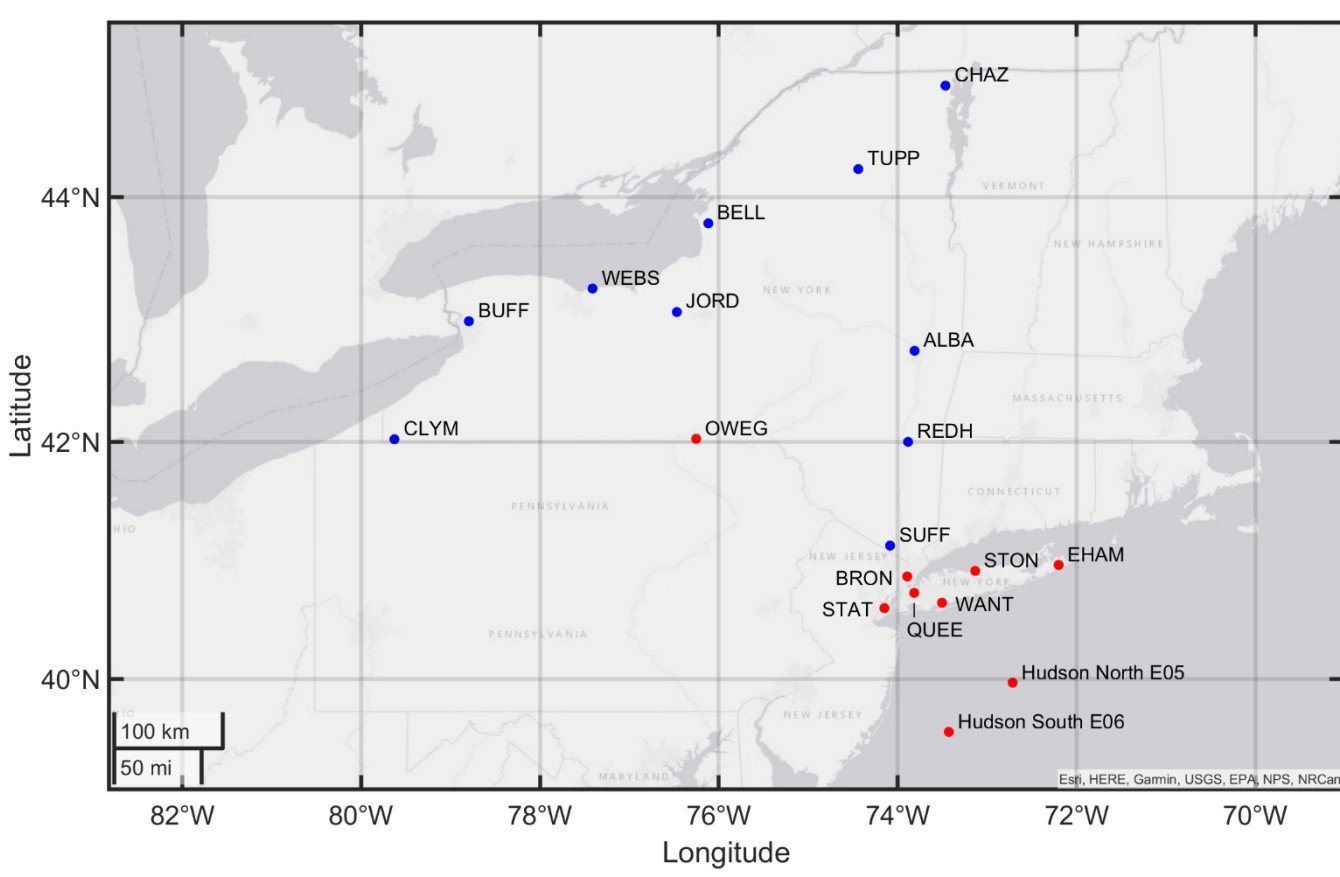

**Figure 1. Map of the locations of the two NYSERDA LiDAR buoys and the 17 NYSM stations. Red points indicate the buoys and seven NYSM sites with the highest data recovery; blue points indicate the remaining NYSM sites.**

## 2.3 ERA5 reanalysis

Wind data from the European Centre for Medium-range Weather Forecasting ERA5 reanalysis is used to provide a
climatological context for analyses of the LiDAR data. Although the LiDAR data sets that we analyze here are – to our
knowledge – unique in terms of the duration and number of sites considered, we also contextualize the results and inferences
drawn from these multi-year, but relatively short duration, observations using the > 40 year duration ERA5 reanalysis product

(Hersbach et al., 2020). This analysis explicitly acknowledges the presence of low-frequency variability (seasonal to multi-decadal) in mid-latitude wind speeds and wind resources (Pryor et al., 2020a) and is designed to quantify the uncertainty on mean wind speeds and power production computed from the relatively short LiDAR data time series.

The ERA5 reanalysis system assimilates a broad range of observing station, buoy, radiosonde, and satellite data and many atmospheric variables including wind components are available at an hourly time step with a spatial resolution of $0.28° \times 0.28°$ (Hersbach et al., 2020). Herein we analyze once-hourly estimates of the u- and v- wind components at a height of 100 m . We use ERA5 output for the period of record with highest quality data assimilated into the reanalysis system: 1979-2022. This time period also includes the observational period of the LiDARs. ERA5 estimates of wind and wave conditions has been extensively independently evaluated and shown to exhibit relatively high fidelity (Pryor et al., 2020b;Gramcianinov et al., 2020;Sharmar and Markina, 2020;Hallgren et al., 2020). However, past research has also indicated substantial spatio-temporal variability in the fidelity of ERA5 wind speed products of relevance to wind energy contexts (Pryor et al., 2020b;Kalverla et al., 2020;Meyer and Gottschall, 2022;Knoop et al., 2020). Here we are using ERA5 output to (i) examine climatological variability and thus contextualize the short observational records, (ii) provide context for the spatial decay of association manifest in the remote sensing observations and (iii) quantify the bias in annual mean wind speeds due to seasonal bias in LiDAR data availability.

## 2.4 Electricity demand

Electrical demand (in MWh) for New York state are also presented and mean values are computed for each hour of the day and each month of the year based on hourly values for 2016-2022 as reported by the U.S. Energy Information Administration (EIA) hourly electric-grid monitor (https://www.eia.gov/electricity/gridmonitor/dashboard/electric_overview/US48/US48).

## 3 Methods

### 3.1 Wind resource and potential power production

Two-parameter Weibull distributions ($A$ = scale and $k$ = shape) are fitted using maximum likelihood estimation (Pryor et al., 2004) and used to describe the probability distributions of wind speeds ($U$) at/close to 150 m height from each LiDAR:

$$f(U) = \frac{k}{A} \cdot (\frac{U}{A})^{k-1} \cdot exp(-(\frac{U}{A})^k) \tag{1}$$

The power in the wind that can be harnessed by wind turbines is often described using the energy density which can be derived from the time series of wind speed measurements or the Weibull distribution parameters:

$$E = \frac{1}{n} \cdot \frac{1}{2} \cdot \rho \cdot \sum_{1}^{n} U^3 = \frac{1}{2} \cdot \rho \cdot A^3 \cdot \Gamma(1 + \frac{3}{k}) \tag{2}$$

where $E$ is in Wm$^{-2}$, $\rho$ is the air density, and $n$ is the number of time stamps from which wind speeds are available and $\Gamma$ is the gamma function (Troen and Lundtang Petersen, 1989).

The electrical power that would be generated by a wind turbine located at each LiDAR site is determined using the power curve from the International Energy Agency (IEA) 15 MW reference wind turbine which has a hub-height of 150 m and a rotor diameter of 240 m (Figure 2). We acknowledge that the physical dimensions and rated capacity of wind turbines deployed offshore are much larger than those that have traditionally been deployed onshore, but use of a single wind turbine allows direct comparison across sites. The time series of 10-minute power production and Annual Energy Production (AEP, in MWh/yr), i.e., the sum of the electrical power production in a year from a single 15 MW wind turbine at each LiDAR location, are used herein for the estimation of electrical power production and power quality.

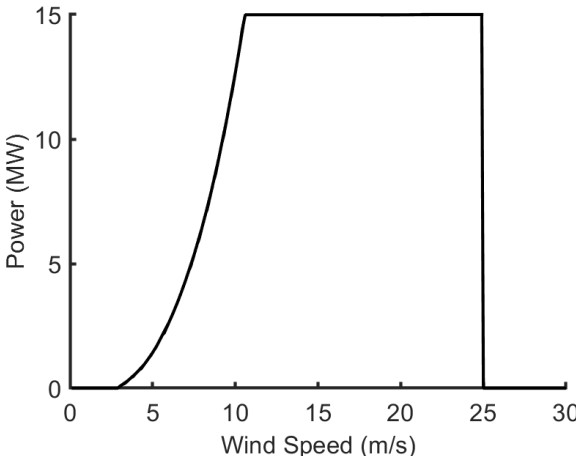

**Figure 2. Power curve for the IEA 15 MW reference wind turbine (Gaertner et al., 2020).**

**3.2    Power quality**

The probability of wind speed and power production ramp events are computed from the NYSERDA and NYSM LiDARs and in the case of wind speeds are normalized as follows:

$$\frac{\delta u(t)}{\sigma_{\delta u}} = \frac{u(t+\tau) - u(t)}{\sigma_{\delta u}} \tag{3}$$

where $u(t)$ is the wind speed at time $t$, $\delta u(t)$ is the wind speed increment from the prior time step, $\tau$ is the chosen time increment, and $\sigma_{\delta u}$ is the standard deviation of the wind speed increments (DeMarco and Basu, 2018). 230 $\frac{\delta u(t)}{\sigma_{\delta u}} = 2$ indicates an increase in wind speed between two consecutive measurements (here $\tau = 10$ minutes) of a magnitude that is equal to two standard deviations of wind speed changes computed from the entire time series, and thus lies in highest 2.5% of values. Conversely, $\frac{\delta u(t)}{\sigma_{\delta u}} = -2$, has a similarly low probability but is associated with a large magnitude decline in wind speed between two consecutive measurements.

Spatial and temporal correlation coefficients are also presented herein. In all cases, non-parametric Spearman rank correlation 235 coefficients are used because wind speeds and power production are not Gaussian distributed variables (Wilks, 2011). Temporal autocorrelation coefficients of the power production time series are used to derive e-folding time scales (i.e. the time delay at which the correlation coefficient drops to $e^{-1}$, i.e. ~ 0.37) which is used to represent the time scale at which the system 'loses' the memory of the initial state (Wilks, 2011). To assess the statistical significance of the correlation coefficients, a student's t-test is used (Wilks, 2011). In this process, a t-statistic is computed from the correlation coefficient ($r$) and the 240 sample size ($n$):

$$t = r \cdot \sqrt{\left(\frac{n-2}{1-r^2}\right)} \tag{4}$$

Due to the high correlation in time, $n$ is corrected to the effective sample size ($n'$) using:

$$n' \approx n \cdot \frac{1-r_1}{1+r_1} \tag{5}$$

where $r_1$ is the lag 1 autocorrelation and $n$ is the total number of samples. The resulting t-score is compared with critical values ($t_{crit}$) for $n'$. If $t > t_{crit}$, the correlation coefficient is statistically different from zero for a confidence level of 99% and the wind speed time series or electrical power production time series from two sites are significantly correlated.

Spatial correlation coefficients are also computed for power production time series from the onshore and offshore sites to examine the association as a function of separation distance and thus the degree to which power production across sites will

be synchronized in time. The e-folding concept can also be applied in this context, to quantify the distance at which the power production from two sites is no longer significantly correlated. Past research has generally found that the correlation between wind speeds and wind power production from wind farms exhibits an exponential decay with increasing separation distance (St. Martin et al., 2015). Herein we fit both single exponential and double exponential fits with the forms:

$$y = a \cdot \exp(b \cdot x) \tag{6a}$$

$$y = a \cdot \exp(b \cdot x) + c \cdot \exp(d \cdot x) \tag{6b}$$

where $y$ is the Spearman correlation coefficients ($r$) for the time series of 10-minute power production estimated for each pair of NYSERDA and NYSM sites, and $x$ is the spherical separation distance between each pair of locations. Fit coefficients; $a$, $b$, $c$ and $d$ are derived using maximum likelihood estimation (Wilks, 2011).

### 3.3 Wind profiles

To quantify the wind shear across the rotor plane we invoke the power law:
$$\frac{U_1}{U_2} = \left(\frac{z_1}{z_2}\right)^{\alpha} \tag{7}$$

where $U_x$ is the wind speed at height ($z_x$) and $\alpha$ is the shear coefficient $f$(stability, surface roughness length) (Irwin, 1979). The International Electrotechnical Commission (IEC) 61400-1 standard states the expected value of $\alpha$ over land is 0.2 and is typically in the range of 0.05 to 0.25 and uses a value of 0.2 in the normal wind profile model (IEC, 2019). The occurrence of $\alpha$ beyond this range implies shear across the rotor plane differs from this design expectation and hence may indicate higher mechanical loading. Profiles of wind speeds from the NYSM LiDARs are used with equation (7) to quantify the frequency of occurrence of anomalous shear in two classes; negative shear exponents ($\alpha < 0$) and $\alpha > 0.3$ conditionally sampled to include only periods when the 150 m wind speed is above 3 ms$^{-1}$, the cut-in for the IEA 15 MW reference wind turbine. Due to the very low frequency of reported wind speeds at 50 m from the NYSM LiDARs, this analysis is performed using wind speeds from 100 m and 250 m, which is sufficient to conform to the IEC standard recommendation that the shear be computed over a height differential of at least one-third of the rotor plane.

To capture LLJ of possible relevance to wind energy applications, LLJ are identified here in any wind speed profile that exhibits a vertically confined wind speed maximum in the lowest 500 m of the atmosphere with wind speeds above and below that level that are $> 2$ ms$^{-1}$ slower than in the maximum (Aird et al., 2021;Aird et al., 2022). This is to ensure the results are comparable to those reported previously for offshore regions of the U.S. east coast that used an analysis vertical window of 20 to 530 m (Aird et al., 2022). Alternative metrics to detect LLJs have been proposed, including use of normalized wind increments by the height interval (i.e. a shear definition) (Hallgren et al., 2023;McCabe and Freedman, 2023).

### 3.4 Climatological context

Hourly zonal ($u$) and meridional ($v$) wind components at 10 and 100 m height are obtained for all ERA5 grid-cells in that contain NYSM and NYSERDA sites, and are converted to wind speed at 150 m height ($U150_{ERA5}$) using $\alpha$ derived from wind speeds at 10 and 100 m computed using equation (7). The mean shear exponent computed from wind speeds at 10 and 100 m height is 0.21, with variability over monthly and interannual timescales of less than 3 %, yielding a multiplier on the 100-m wind speed of 1.09 which is applied to obtain $U150_{ERA5}$. Hourly $U150_{ERA5}$ estimates are used to calculate hourly wind production ($P150_{ERA5}$) using the IEA 15 MW reference wind turbine. The long-term records of $U150_{ERA5}$ and $P150_{ERA5}$ are used to assess the uncertainty in annual mean wind speeds and AEP resulting from the limited duration data records at the NYSM and NYSERDA LiDARs using a bootstrapping approach (Wilks, 2011). Hourly values from the 40-year $U150_{ERA5}$ and $P150_{ERA5}$ record are randomly resampled 1000 times with replacement using the number of hours from each month that the LiDAR data are available (Figure 3). For each of these 1000 bootstrapped samples the annual mean wind speed and AEP is

calculated to provide an estimate of uncertainty that arises due to the short time series from the LiDARs. Additionally, Spearman correlation coefficients between the time series of $P150_{ERA5}$ at all NYSM and NYSERDA grid-cells are calculated for the full 44-year record and used to contextualize the spatial correlation derived using the LiDAR measurements.

## 3.5    LCoE

As indicated above, there are many possible advantages in deploying wind turbines offshore as a component of the electricity generation system. One potential disadvantage is that offshore wind energy generation costs are expected to be higher than those from onshore wind, although still less than those from nuclear (Barthelmie et al., 2023). The simple Levelized Cost of Energy (LCoE) model applied here is similar to that developed in Barthelmie et al. (2023):

$$LCoE = \frac{CAPEX \cdot CRF + OPEX}{AEP} \tag{8}$$

where: *CAPEX* is the capital costs, *CRF* is the cost recovery factor, and *OPEX* is the annual operations and maintenance. Fixed costs are used here (Table 1), and *AEP* is the Annual Electricity Production from analyses described herein. Project lifetimes are assumed to be 30 years and no adjustment is made for turbine availability or other losses such as wakes or electrical losses. For the offshore locations, CAPEX is calculated from the values in Table 1 with no adjustments for distance to the coast, water depth, etc., and the water depth is appropriate for bottom-mounted wind turbines. Thus, the estimated LCoE from this simplified model are best case values.

**Table 1. Key parameters for the LCoE model. Values taken from: (Stehly and Duffy, 2022).**

|  | Onshore | Offshore |
|---|---|---|
| Capital expenditures (CAPEX) (million$/MW) | 1.501 | 3.871 |
| • Turbine (million$/MW) | 1.03 | 1.3 |
| • Fixed charge rate i.e. cost recovery factor (CRF) (%) | 5.88 | 5.82 |
| • Project costs (million$/MW) | 0.120 | 0.67 |
| • Foundation (million$/MW) | 0.075 | 0.496 |
| • Electric infrastructure including sub-stations (million$/MW) | 0.132 | 0.693 |
| • Finance ($/MW) (plus other costs for offshore includes e.g., decommissioning) | 0.113 | 0.704 |
| Operational expenditures (OPEX) ($/MW/yr) | 0.04 | 0.111 |

## 4    Results

### 4.1    Wind resource, potential power production, and LCoE

Wind speed time series from all the LiDARs (Table 2) indicate similar seasonality due to their relative proximity. Highest monthly mean wind speeds at ~150 m occur during the cold season (November to March) and lowest values are observed during summer (July and August) (Figure 3). This is consistent with the climatology of the U.S. northeast with the cold season months exhibiting a high frequency of mid-latitude cyclone passages and with data from operating wind farms that exhibit highest CF during late winter and early spring (Pryor et al., 2023). The data also indicate considerably higher wind speeds at 150 m based on data from the LiDARs deployed offshore (Figure 3). The mean wind speeds at this height from the two NYSERDA buoy-mounted LiDARs are 10.1 ms$^{-1}$, while the mean wind speed from the Owego NYSM site (located < 400 km away) is 7.72 ms$^{-1}$ (Table 2). In August, the mean monthly wind speed at the Hudson North buoy is 7.76 ms$^{-1}$ and at Owego is 6.08 ms$^{-1}$; in December, the mean monthly wind speed at these two sites is 11.24 ms$^{-1}$ and 9.16 ms$^{-1}$, respectively. Data from the LiDAR buoys also show a consistently higher frequency of $U = 15$-$25$ ms$^{-1}$ when the IEA 15 MW reference wind turbine would operate at rated capacity (Figure 2). Figure 3 further indicates the presence of seasonality in data availability. The excess representation of August in the Hudson North E05 data will tend to lead to a negative bias in the overall wind resource and estimated power production because wind speeds in that month are typically lower than other months (Figure 3). The mean

monthly wind speed from August-November in data from the Hudson South E06 LiDAR is 9.75 ms$^{-1}$, which is below the overall mean, so the relatively low data availability in these months at E06 may also lead to a small negative bias in the derived
315    mean energy density and power production (Figure 3). The seasonality in data availability is particularly consistent and amplified at the NYSM sites. As shown in Figure 3, at WANT (the site with the highest seasonal bias in data availability) over 12% of the total observations were recorded in July while in a data set free of availability bias this value would be 8.5%. Bootstrapping of ERA5 data indicates the mean annual wind speed computed from the LiDAR time series at the NYSM sites is likely underestimated by ~ 1.5-4.5% while AEP is underestimated by ~3-10% due to the high data availability in summer.
320    Analyses of wind speed data from the NYSM LiDARs at all measurement heights from 100 to 500 m indicates that, averaged across all stations, the data availability as a function of height varies only by ±2.5%.

**Table 2. Weibull distribution parameters from the 150 m wind speed time series (and 95% confidence intervals, CI) and energy density derived from those parameters. AEP computed using the IEA 15 MW wind turbine power curve, along with the frequency of zero power and power production at rated. The data shown in italics are computed using the most complete continuous 12-month period. The column headed e-folding time shows the time for the Spearman correlation coefficient to fall below $e^{-1}$. The ninth and tenth columns show the frequency of extreme shear for all periods when $U$ at 150 m > 3 ms$^{-1}$. The following column shows estimated Levelized Cost of Energy (LCoE) values derived using the assumptions described in section 3.5 based on AEP estimates shown in the fifth column and derived using the LiDAR observations.**

| Site | Weibull Scale Parameter (A) (m/s) [CI] | Weibull Shape Parameter (k) [CI] | Energy Density (E) (W/m²) | AEP (GWh/yr) | Frequency of no power production: $U$ < 3 / $U$ > 25 ms$^{-1}$ (%) | Frequency of maximum (rated) power production (%) | e-folding time (hr) | Frequency of $\alpha$ < 0 | Freq of $\alpha$ > 0.3 | LCoE ($/MWh) |
|---|---|---|---|---|---|---|---|---|---|---|
| BRON | 6.915 [6.896, 6.935] | 2.013 [2.005, 2.021] | 267 | 35.6 | 15.3/0.06 | 5.16 | 8.0 | 15.0 | 23.4 | 49.0 |
|  | *7.146* | *2.031* | *293* | *38.7* | *14.8/0.07* | *5.67* | *8.0* |  |  |  |
| EHAM | 10.16 [10.13, 10.18] | 2.153 [2.144, 2.161] | 794 | 71.7 | 6.38/0.31 | 21.2 | 9.3 | 15.4 | 17.0 | 26.0 |
|  | *10.32* | *2.202* | *816* | *73.0* | *5.63/0.40* | *23.1* | *9.3* |  |  |  |
| OWEG | 8.703 [8.678, 8.727] | 2.172 [2.163, 2.181] | 496 | 58.3 | 9.02/0.18 | 11.4 | 9.7 | 10.9 | 13.9 | 33.9 |
|  | *8.514* | *2.168* | *465* | *56.0* | *8.96/0.19* | *12.5* | *9.0* |  |  |  |
| QUEE | 7.607 [7.587, 7.627] | 2.000 [1.993, 2.008] | 358 | 43.3 | 12.2/0.14 | 8.66 | 8.8 | 19.0 | 17.3 | 45.3 |
|  | *7.484* | *2.009* | *340* | *41.9* | *13.0/0.12* | *7.75* | *7.8* |  |  |  |
| STAT | 7.602 [7.580, 7.625] | 2.014 [2.006, 2.022] | 355 | 43.6 | 12.4/0.14 | 6.49 | 7.2 | 15.9 | 20.2 | 43.7 |
|  | *7.550* | *2.029* | *345* | *43.4* | *13.0/0.12* | *6.17* | *7.3* |  |  |  |
| STON | 9.423 [9.396, 9.450] | 2.039 [2.030, 2.048] | 668 | 64.1 | 9.29/0.11 | 15.6 | 9.7 | 16.7 | 15.2 | 29.4 |
|  | *9.447* | *2.037* | *674* | *64.4* | *9.29/0.13* | *17.5* | *9.8* |  |  |  |

| | | | | | | | | | |
|---|---|---|---|---|---|---|---|---|---|
| WANT | 9.282 [9.254, 9.309] | 1.970 [1.962, 1.977] | 662 | 60.0 | 7.64/0.47 | 13.8 | 8.3 | 25.3 | 10.4 | 31.6 |
| | *9.185* | *2.011* | *627* | *60.1* | *8.01/0.45* | *17.1* | *7.8* | | | |
| Hudson North E05 | 11.40 [11.36, 11.43] | 2.127 [2.117, 2.137] | 1134 | 79.7 | 5.30/0.48 | 30.6 | 11.3 | | | 61.9 |
| | *11.42* | *2.147* | *1130* | *80.5* | *5.48/0.53* | *36.3* | *11.7* | | | |
| Hudson South E06 | 11.38 [11.34, 11.42] | 2.123 [2.111, 2.134] | 1131 | 80.0 | 5.94/0.46 | 27.2 | 10.0 | | | 64.4 |
| | *11.04* | *2.073* | *1056* | *77.4* | *6.84/0.31* | *21.4* | *10.7* | | | |

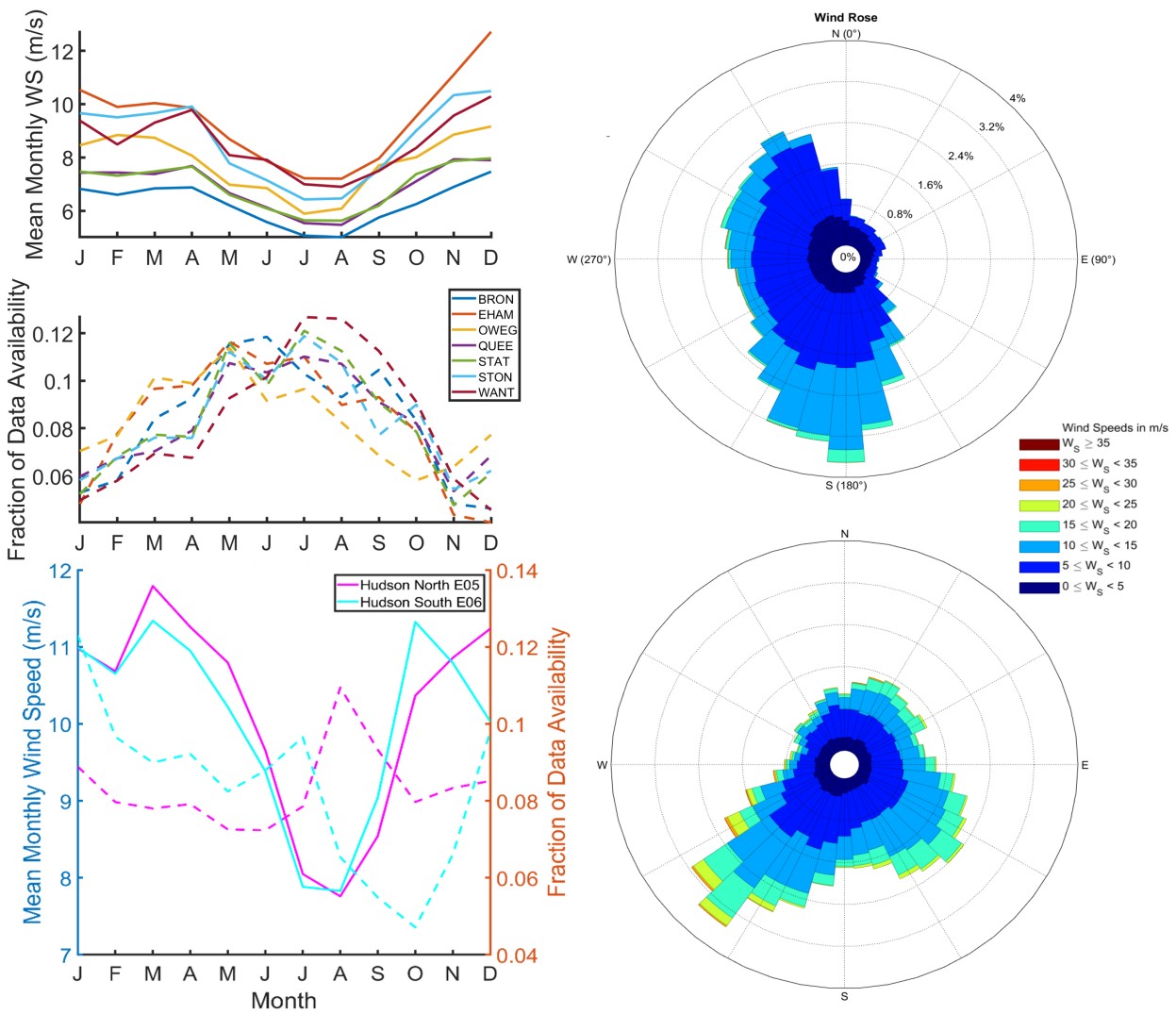

**Figure 3. Monthly mean wind speed at ~150 m (ms⁻¹, solid lines) and fraction of data availability (dashed lines) for the seven NYSM sites with highest data availability (top/middle left) and both NYSERDA buoy sites (bottom left), as well as wind roses for the OWEG (top right) and Hudson North E05 (bottom right) sites. A value of 0.08 for the fraction of data availability for a given month indicates 8% of the total sample is comprised of values recorded in that month.**

The Weibull distribution fits to 150 m wind speeds from the buoy-mounted LiDARs have very similar shape and scale parameters (Figure 4 and Table 2). Consistent with expectations, the Weibull scale parameters from the NYSERDA buoys are also substantially higher than those from the seven NYSM sites and exceed values from the NYSM by 2 ms⁻¹ for all sites except EHAM which is on Long Island and within 1 km of the coastline (Figure 1). The Weibull distribution parameters translate to higher energy densities at the locations of the buoys (Table 2). This is also true for calculations based on the 'best year' of data (Table 2). When wind speeds from the 'best year' are used to compute the Weibull fits and AEP, differences of 0.1-3% in the Weibull scale parameters and 1-8% in AEP are found relative to estimates from the longest available records (Table 2). Even compared to the NYSM location with the highest Weibull scale parameter and highest mean wind speeds (EHAM), both buoys have over 40% higher energy density. Application of the power curve from the IEA 15 MW reference turbine to the wind speed time series yields AEP values for the buoy-mounted LiDARs that are a factor of almost three higher than some of the NYSM sites (e.g., QUEE and STAT) and nearly twice as much as many NYSM stations except EHAM (Table 2). Thus, consistent with expectations, the wind speed time series from the LiDARs operated on the NYSERDA

buoys indicate a substantially better wind resource and higher projected electrical power output (AEP) than is estimated based on data from the NYSM LiDARs.

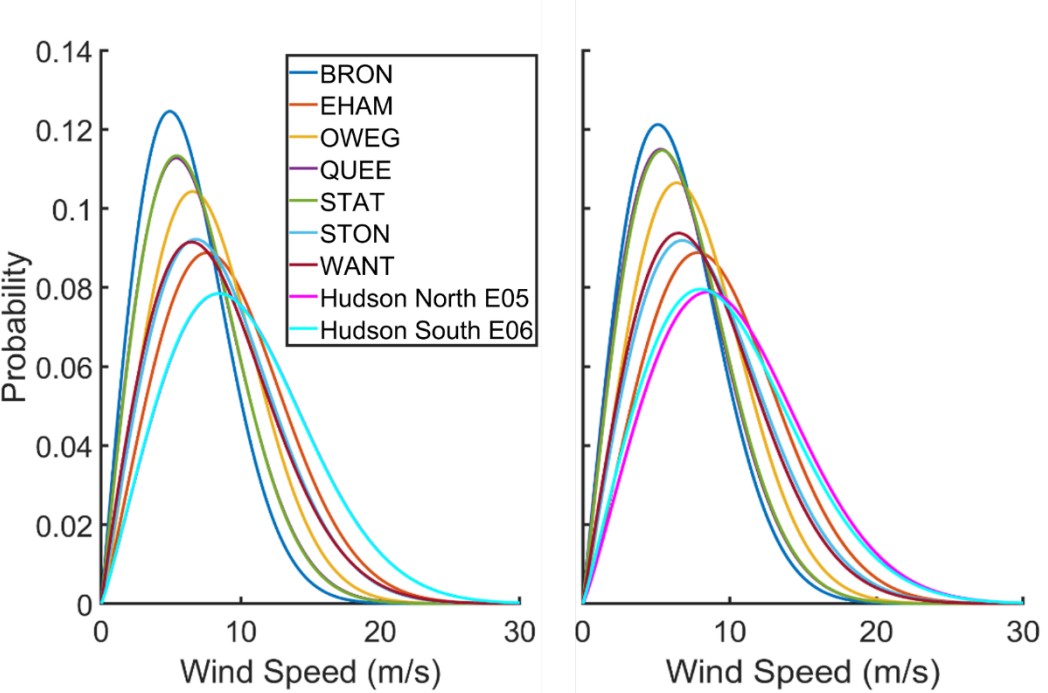

**Figure 4. Probability distributions from the Weibull fits to 10-minute wind speeds at 150 m height from the NYSERDA LiDAR buoys and the NYSM stations for all available data (left) and year with common highest data availability ('best year' of September 2018-2019) (right). Note: probability distributions from Hudson North E05 and Hudson South E06 virtually overlay each other in the left panel.**

Despite higher projected AEP for the offshore locations, the additional costs involved in installing and operating offshore wind

farms results in higher LCoE estimates for the offshore sites (Table 2). LCoE estimates derived using AEP at the NYSM sites and assumptions stated in section 3.5 (Table 1) are 26 to 49 $/MWh while estimates for the NYSERDA buoy locations are 62-64 $/MWh.

### 4.2 Power quality

Three aspects of power quality are evaluated using the wind speed at ~150 m ($U$) and power production time series. The first

is the probability of wind speeds at which no power is produced; $U < 3$ ms$^{-1}$ or $U > 25$ ms$^{-1}$. The probabilities of wind speeds below the cut-in speed of the IEA 15 MW wind turbine ($U < 3$ ms$^{-1}$) are substantially higher for the NYSM sites than the offshore locations (Table 2 and Figure 4). Indeed, for three of the seven NYSM sites the probability of wind speeds below cut-in is well over twice that for the offshore sites, and even the locations of Long Island that are very close to the coast (EHAM and WANT) exhibit considerably higher frequency of $U < 3$ ms$^{-1}$ than is derived using data from NYSERDA LiDARs (7.6

and 6.4% versus 5.3 and 5.9%, see Table 2). The frequency of $U$ above cut-out ($U > 25$ ms$^{-1}$) is higher based on data from the LiDARs on the buoys, but the overall frequency is low at all locations (< 0.5%). Thus, wind turbines deployed offshore at the NYSERDA buoy locations will produce some power on a considerably larger fraction of the time than any of the onshore locations. This inference is true whether the entire time series or the "best year" of data are considered.

The second component of power quality is the intermittency in terms of the probability and magnitude of ramp events – that

is rapid changes in wind speed and/or power production. Wind speed time series at 150 m height from the NYSM and NYSERDA LiDARs indicate clear similarities in terms of ramp event magnitude and frequency to those derived using data from the FINO1 platform in the North Sea, Cabauw onshore in western portion of the Netherlands, Høvsøre in coastal Jutland, Denmark, and NWTC in the foothills of the Colorado Rocky Mountains (DeMarco and Basu, 2018) (Figure 5). Data from the

NYSERDA buoys indicate a low probability of wind speed ramps of all magnitudes relative to the NYSM LiDARs (Figure
5), and all LiDAR time series indicate a substantially higher probability of a ramp-up (increase) than a ramp-down (decrease)
of a given magnitude in wind speeds. Wind speed ramps in hourly ERA5 data exhibit a narrower distribution owing to spatial
and temporal averaging, illustrating the need for in-situ data for capturing high resolution wind variability (Figure 5).
Consistent with the lower probability of large-magnitude rapid changes in wind speed offshore, data from the NYSERDA
buoys (Hudson North E05 and Hudson South E06) indicate probabilities of a wind power ramp with $> \pm20\%$ change in power
are considerably lower than those from any of the onshore locations (Figure 5). Thus, the chance of experiencing an increase
or decrease in electrical power production of 20% from one 10-minute period to the next is substantially lower for wind
turbines deployed offshore. This indicates that wind turbines deployed offshore are likely to exhibit less intermittency in terms
of electrical power production which is critical to efficient grid integration (Ayodele et al., 2012).

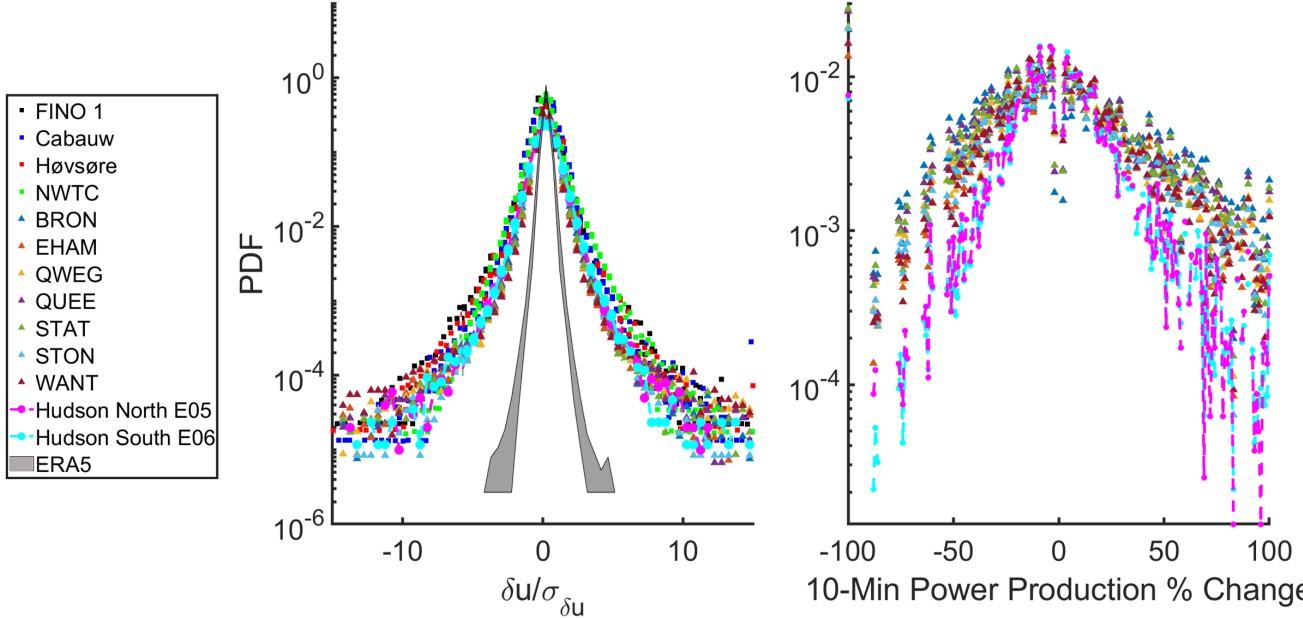

**Figure 5. Left: probabilities of wind speed ramp events computed from the 10-minute data from the NYSERDA LiDAR buoys and
the NYSM sites computed using Equation (3), and reported for four locations at or near operating wind turbines: FINO 1 is
(offshore) in the North Sea, Cabauw is in the western portion of the Netherlands, Høvsøre is in Jutland, Denmark, and NWTC is in
the foothills of the Colorado Rocky Mountains (data digitized from: (DeMarco and Basu, 2018)). Wind ramps computed from the
hourly ERA5 output are shown by the gray polygon. Right: probabilities of wind power production ramp events at the locations of**
**the NYSERDA buoys and the NYSM sites computed by applying the power curve for the IEA 15 MW reference wind turbine to the
LiDAR wind speeds. The probabilities of no-change (i.e., power ± 0%) are not shown to aid visibility.**

The third aspect of power quality is predictability. The autocorrelation in power production at different time lags for the NYSM
LiDARs exhibit clear diurnal oscillations and shorter e-folding time scales. Power production estimates using wind speeds
from the Hudson North E05 buoy show the largest e-folding time of ~68, 10-minute periods (11.3 hours), and ~70, 10-minute
periods (11.7 hours) in the 'best year' of data. Comparable estimates for data from the Hudson South E06 buoy are ~ 60 and
64, 10-minute periods (10.0 and 10.7 hours) (Table 2 and Figure 6). These relatively large e-folding times for the buoy locations
indicate a longer atmospheric 'memory' at these sites, indicating the potential for more accurate short-term power prediction
forecasts because each time step is strongly dependent on the value in previous time step(s).

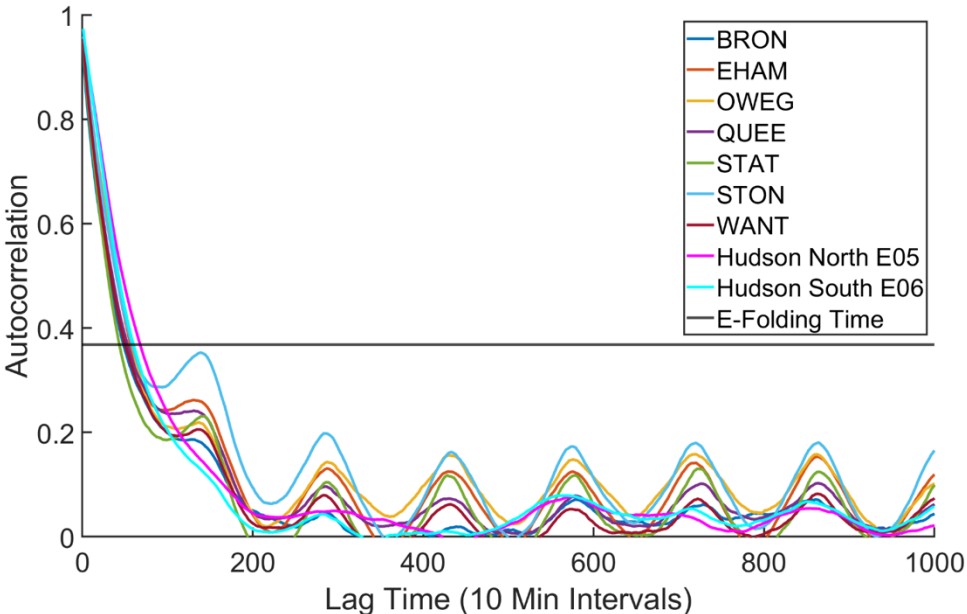

**Figure 6. Temporal autocorrelation (computed using Spearman correlation coefficients) of the wind power production at different lag times based on data from the NYSERDA buoys and the NYSM stations. The horizontal line denotes a correlation of e$^{-1}$ which is used here as a first order estimate of the e-folding time.**

### 4.3 Spatial correlation

While power production from wind farms is inherently intermittent at the local scale, aggregation over large spatial scales
reduces power fluctuations (Potisomporn and Vogel, 2022;Pryor et al., 2014;Simão et al., 2017;Pryor et al., 2020b;St. Martin et al., 2015). However, the optimal spatial scale of integration is likely to be a strong function of the prevailing meteorology. Thus, an analysis of power production computed based on LiDAR data at each of the NYSM profiler stations and NYSERDA buoys is undertaken to quantify the spatial decorrelation scale. Consistent with the a priori expectation based on past research, the correlation of time-series of estimated power production at the different locations decays exponentially with increasing
separation distance (Figure 7). The highest correlation coefficient is between power production time series from the two NYSERDA buoys (0.834, see SM Table 1 and Figure 7). NYSM sites EHAM and STON have an almost identical separation distance as the buoys (Figure 1), but these time series of estimated power production have a slightly lower correlation coefficient (0.764) due to variability caused by the presence of land use land cover and terrain features onshore. For the sample sizes of data from the LiDARs and a lag-1 autocorrelation of > 0.9, application of equations (4) and (5) imply the power
production time series would be considered fully de-correlated at Spearman correlation coefficients < 0.2. As shown in Figure 7 this level is not reached for the sites at which the LiDAR are deployed. Nevertheless, exponential fits to correlation coefficients as a function of separation distance imply that on average the correlation coefficients drop below about 0.4 for separation distances of ~ 350 km. This suggests that careful siting of wind farms on- and off- shore could be used to decrease coherent variations in electrical power production within the NY ISO. Output from ERA5 when converted to electrical power
production exhibits higher correlation coefficients at similar separation distances to the LiDARs consistent with the higher spatial smoothing inherent in reanalysis products (Figure 7).

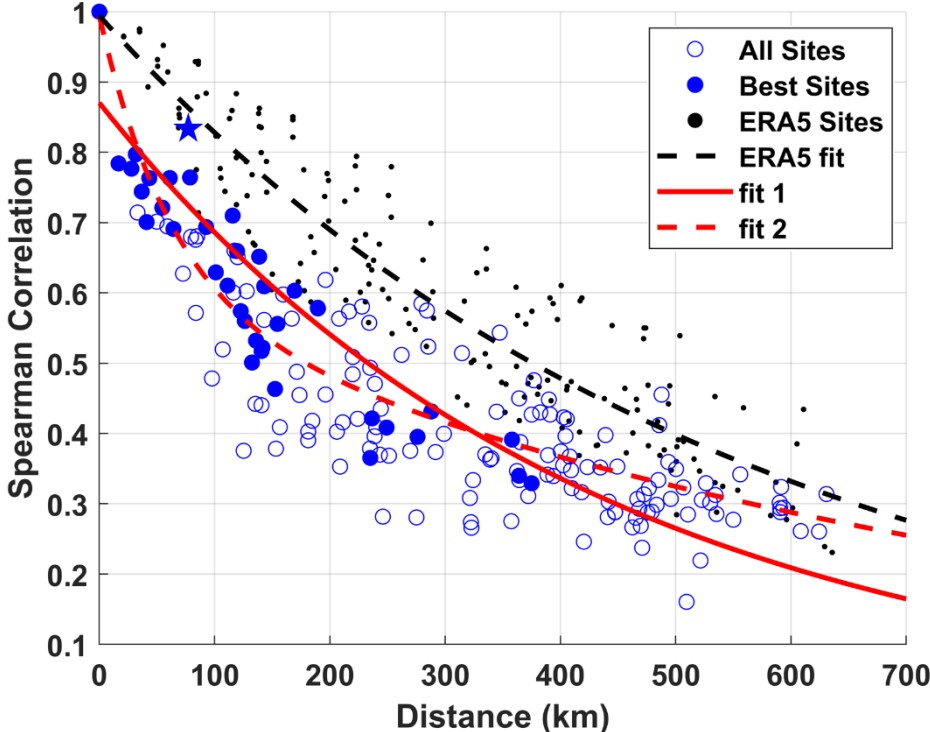

**Figure 7. Spearman spatial correlation coefficient (*r*) of power production for the LiDAR (blue) and ERA5 (black) output sampled at the NYSERDA and NYSM locations against the separation distance between all 19 sites (17 onshore, 2 offshore). Solid blue points indicate location pairs with high data availability, open points represent the relationships including the remaining NYSM sites. The star represents the correlation between the two NYSERDA buoys. Best fit lines (y = Spearman correlation coefficient, x = spherical distance between locations) are shown for the LiDAR: y = 0.8703exp(-0.002377x) (solid red) and y = 0.4021exp(-0.01595x) + 0.5914exp(-0.0012x) (dashed red). The fit to the ERA5 estimates has the form y = 0.9942exp(-0.00183x) (dashed black).**

### 4.4    Shear conditions and LLJ at the NYSM onshore sites

LiDAR data from all 17 NYSM sites indicate a very high frequency of extreme wind shear (Table 2). This is likely due in part to the heights being considered lying outside of the surface layer when the wind power law is most likely to be an appropriate approximation. Nevertheless, all NYSM LiDARs have a very high frequency of shear exponents computed using wind speeds at 100 and 250 m for wind speeds at 150 m of 3 to 25 ms⁻¹ that lie beyond the 0 to 0.3 expected range. At all the NYSM sites, 5% of shear exponent values during wind turbine operation lie above 0.39 and a further 5% of values fall below -0.09. The

high frequency of extreme positive shear at many of the sites is likely to be due to the high surface roughness lengths since many of the NYSM sites are in the southeast of the state in highly urbanized locations. The frequency of negative shear is highest at WANT, on Long Island, likely in part because of the local land use land cover variability. The occurrence of negative shear 10.9-25.0% of the time from the NYSM sites is broadly comparable to the frequency of occurrence of negative shear between heights of 42-292 m (12%) found in WRF simulations over the U.S. state of Iowa (Barthelmie et al., 2020). A high

positive shear exponents ($\alpha > 0.2$) was also found in analyses of WRF output in Iowa (>38%) again consistent with the estimated probability of occurrence derived using the NYSM LiDAR data (100 to 250 m) (Table 2). The implication is that large wind turbines deployed in these locations may experience a relatively high frequency of large unbalanced rotor loads and reduced component lifetimes unless such loads can be appropriately compensated (Hur et al., 2017).

Consistent with the lower wind speeds during the summer (Figure 3), weaker synoptic forcing during this season, and previous

analyses of LLJ offshore (Aird et al., 2022), all NYSM sites exhibit the highest frequency of LLJ occurrence in the summer months (Figure 8). The highest frequency of occurrence (14% of all 10-minute periods) of LLJ occurs during June at EHAM on the coast of Long Island (Figure 8). Analyses of the WRF simulations for this location found a LLJ frequency during June of 11% and a very similar seasonal cycle of occurrence (Aird et al., 2022). The site-to-site variability in LLJ probability at the different NYSM locations is due to local site conditions (e.g., proximity to the coastline, topographic variability and land use

land cover variability) that are linked to the dynamical causes of LLJ (Balsley et al., 2003;Kallistratova et al., 2009;Blackadar, 1957;Holton, 1967). LLJ core heights are also lower during the summer months, with LiDAR observations from WANT indicating a mean LLJ core height of < 280 m during June (Figure 8). However, for most of the NYSM locations the mean LLJ core heights are above 300 m and thus above the swept area even of the IEA 15 MW reference wind turbine. There is a higher probability of LLJ intersecting with the rotor plane during summer. However, LLJ diagnosed from the onshore LiDARs are typically at greater elevations than are indicated offshore by the WRF simulations, where LLJ cores were frequently < 200 m above the sea surface (Aird et al., 2022). It is important to acknowledge that comparisons of LLJ climates derived from LiDAR measurements and WRF modelling should be done cautiously and that LLJ detection from the LiDAR wind speed profiles is critically dependent on unbiased data availability. Nevertheless, this analysis suggests LLJ within the rotor plane, as a source of large, unbalanced rotor loads and reduced blade lifetimes, are less frequent at these onshore locations.

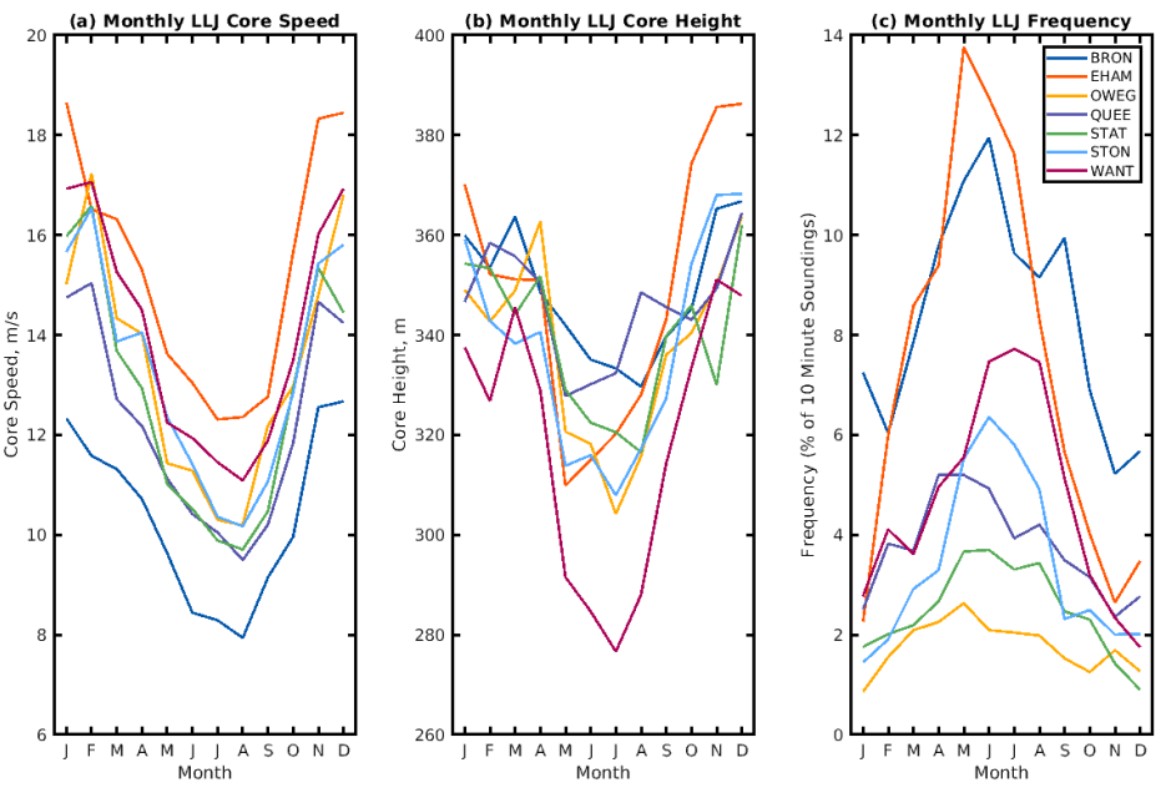

**Figure 8. Monthly mean low-level jet (a) core wind speed (ms⁻¹), (b) core height (m), and (c) the mean frequency of occurrence (probability of occurrence in any given 10-minute LiDAR profile) across the seven NYSM sites. A LLJ frequency of 5% calculated from the LiDAR deployed at QUEE for the calendar month of May indicates that LLJ were indicated in 5% of all 10-minute periods during this month. Data in panels (a) and (b) indicate that at that site in the month of May the associated LLJ mean core wind speed is 11 ms⁻¹ and the mean core height above ground is 330 m.**

## 4.5    Demand matching

Electricity demand in New York state tends to peak in the afternoon (~ 1700 eastern standard time, EST) and in summer (highest values in July), though a secondary maximum occurs in January (Figure 9). Wind power production calculated from the NYSM/NYSERDA LiDARs and ERA5 grid-cell data ($P150_{ERA5}$) show highest values at night (0100 to 0500 EST) and during winter to spring (December-April), with the lowest production during the day (1300 to 1600 EST) and during the summer (Figure 9). Wind power production estimated based on LiDAR data from the NYSERDA buoy locations exhibits markedly lower diurnal and seasonal variability than is estimated at the NYSM sites, varying by ± 10% around the mean versus ± 25% at NYSM. This results in a reduction in mean absolute error (MAE) between time series of normalized WPP from the offshore LiDAR and electricity demand on both diurnal and seasonal timescales. The MAE computed from the mean hourly offshore WPP and demand is 0.19 when computed over the 24 hours of the day (Figure 9a) and 0.13 when computed from the time series of monthly mean values (Figure 9b). Both are smaller than MAE computed from WPP from the onshore LiDARs

and demand on these time scales which are 0.25 and 0.20, respectively. This implies there will be better matching to electricity demand for power production from wind turbines deployed offshore.

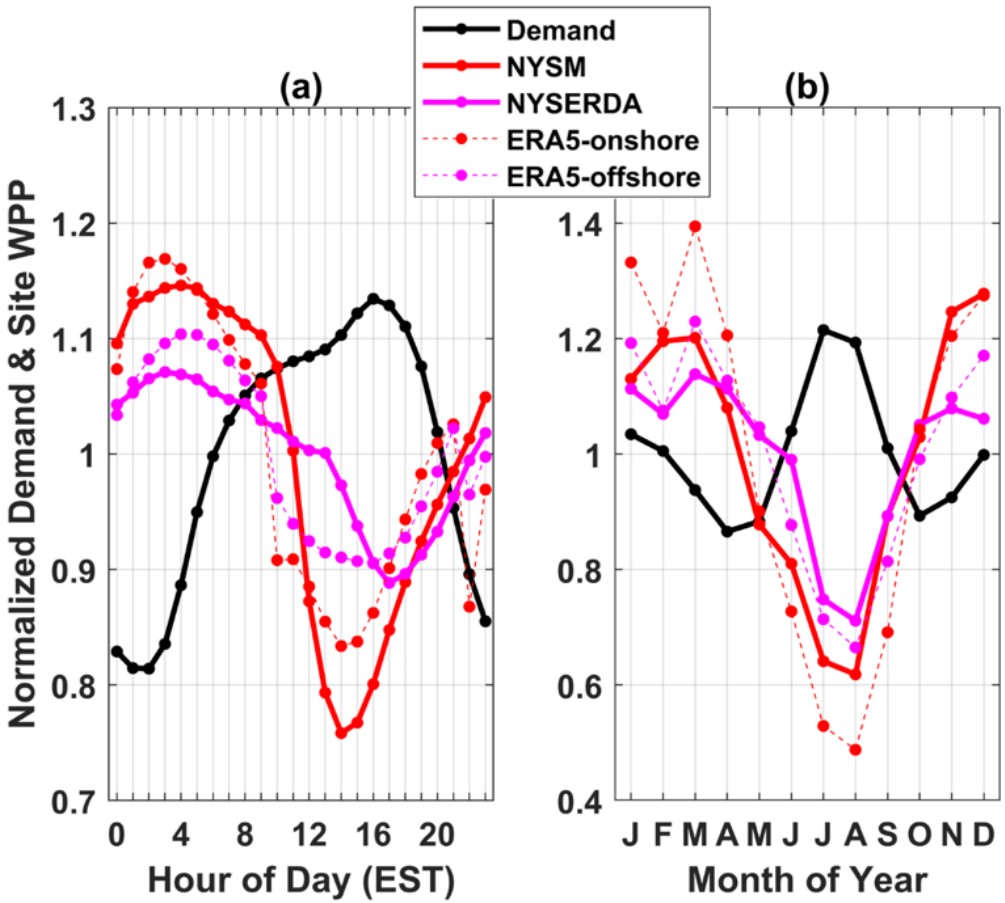

**Figure 9. Normalized (a) diurnal and (b) monthly cycles of electricity demand for New York State (black) and wind power production (WPP) at NYSM (red) and NYSERDA (magenta) sites. ERA5-derived WPP is shown for the grid-cells which contain the NYSM and NYSERDA LiDARs (thin, dashed lines) for the climatological period, 1979-2022. The data are normalized to a mean value of 1 and so that values of 0.9 or 1.1 in a given hour or month indicates WPP or demand that is 10% below or above the mean, respectively.**

## 5    Concluding remarks

Comparative analyses of wind resources and projected power production quantity and quality at onshore and offshore locations have been hampered by the lack of high-quality hub-height wind speed observations. Here we use uniquely detailed LiDAR measurements from an onshore profiler network and offshore campaign to compare projections of potential power generation quantity and quality from offshore and onshore locations in New York State (Figure 1). Returning to the objectives articulated in section 1, the study results indicate there are significant benefits to offshore deployments of wind turbines:

• Wind resources at locations in the New York Bight (coastal offshore areas southeast of New York state, Figure 1) greatly exceed those of all onshore locations within New York state. The mean wind speeds at ~ 150 m ($\bar{U}$) offshore are above 10 ms$^{-1}$, while $\bar{U}$ is below 8 ms$^{-1}$ at all onshore sites. Weibull distribution fits to the 10-minute wind speed time series indicate scale parameters that are higher by 2 ms$^{-1}$ than all onshore locations (Figure 4) except EHAM which is on Long Island and is within 1 km of the coastline (Figure 1). Accordingly, energy densities are 40% higher

offshore and power production estimated offshore using the power curve of the IEA 15 MW wind turbine (Figure 2) yield over twice the AEP estimated for all onshore sites except EHAM (Table 2). Power generation estimated from wind speed time series offshore also exhibits lower variability on diurnal and seasonal time scales (Figure 6) and improved matching to current electricity demand in New York state (Figure 9). This implies that not only is the offshore resource considerably larger offshore, but the ability to meet electricity demand is better for wind turbines

deployed offshore. The differences in wind climates, energy density and estimated power production from the offshore and onshore LiDAR are of sufficient magnitude that they likely exceed any discrepancy due to application of different CNR thresholds in data screening proceedures for the two LiDAR networks.

- Analyses presented herein also suggest that power generation intermittency is lower for the offshore sites. The probability of wind speeds below cut-in or above cut-out for the IEA reference wind turbine is lower offshore, as is

the probability of large magnitude wind speed and power ramps (Figure 5). For example, the probabilities of wind power ramps with $> \pm 20\%$ change in power over a 10-minute period are less than half as probable offshore as onshore. The higher temporal autocorrelation of wind power production offshore (Figure 6 and Table 2) may also aid the accuracy of short-term wind power forecasting for wind turbines deployed offshore, yielding economic benefits to wind farm owner/operators and enabling grid integration.

Conversely, the frequency of anomalous wind speed shear and LLJ close to, or within, the rotor plane computed from the NYSM LiDAR wind speed profiles are slightly higher than those previously reported for the offshore areas from numerical simulations (Aird et al., 2022) but LLJ also exhibit higher elevations of the jet cores (Figure 8) and thus may be of less concern to wind turbine loading.

An analysis of the distance dependence of the co-variability of power production derived from measured 10-minute mean wind

speed time series at the onshore and offshore sites indicates that the non-parametric Spearman correlation coefficient drops below 0.4 at distances of about 350 km (Figure 7). This implies that in order to ensure consistency of electrical power production from wind farms in New York state, major developments should be separated by more than 350 km. This information could be used to guide judicious selection of wind farm locations to minimize the probability of concurrent low generation from onshore and offshore sites.

Thus, in accord with a priori expectations, analyses presented herein indicate there are advantages to the emerging trend towards offshore wind energy deployments in terms of the wind resource and the expected power quality and predictability (reduced ramp events, higher probability of rated power, etc.). Despite higher project AEP for the offshore locations, the additional costs involved in installing and operating offshore wind farms results in higher LCoE estimates for the offshore sites (Table 2). LCoE estimates derived using AEP at the NYSM sites are 26 to 49 $/MWh while estimates for the NYSERDA

buoy locations are 62-64 $/MWh. Nevertheless, projected LCoE from wind energy for all of the sites investigated here in NY are competitive with all other electricity generation sources, with the possible exception of utility-scale PV, and much less expensive than traditional sources such as coal and nuclear that, according to a recent analysis, have an unsubsidized LCoE of 65-152 $/MWh and 131-204 $/MWh, respectively (Lazard, 2023).

## 6      Code availability

Analyses presented here were performed using normal functions within MATLAB$^{TM}$. No specialized codes were developed or employed.

## 7      Data availability

The LiDAR data from the NYSERDA buoy campaign are available from: NYSERDA (2022). *E05 Hudson North 10 Minute*. Det Norske Veritas. and NYSERDA (2022). *E06 Hudson South 10 Minute*. Det Norske Veritas. Both data sets retrieved

February 13, 2022, from https://oswbuoysny.resourcepanorama.dnv.com/. Reports documenting LiDAR performance verification are also available for download from: https://oswbuoysny.resourcepanorama.dnv.com/. Specifications for the 15 MW reference wind turbine are available from *GitHub - IEA Wind Task 37/IEA-15-240-RWT*. https://github.com/IEAWindTask37/IEA-15-240-RWT. Data from the New York State Mesonet can be requested from: http://www.nysmesonet.org/. A readme documenting data processing for this network is available from:

## 8    Author contribution

SCP conceptualized the study, managed the project and acquired the data sets used in the study. SCP and RJB acquired the funding and the computing resources used. RF, SCP, JCC, RJB and JAA performed data analyses. RF and JCC led the visualization. RF and SCP wrote the initial draft. All authors contributed to refinement of the final manuscript.

## 9    Competing interests

RJB is a member of the editorial board of Wind Energy Science. The peer-review process was guided by an independent editor, and the authors have also no other competing interests to declare.

## 10    Acknowledgements

The authors are grateful for funding from the U.S. Department of Energy Office of Science (DE-SC0016605), the U.S. Department of Energy Office of Energy Efficiency and Renewable Energy, and the New York State Energy Research and Development Authority via the National Offshore Wind Research and Development consortium (147505 and 147506). This research was further enabled by computational resources supported by the U.S. National Science Foundation via the Extreme Science and Engineering Discovery Environment (XSEDE) (award TG-ATM170024). This research was made possible in part by the New York State (NYS) Mesonet. Original funding for the NYS Mesonet was provided by Federal Emergency Management Agency grant FEMA-4085-DR-NY, with the continued support of the NYS Division of Homeland Security & Emergency Services; the State of New York; the Research Foundation for the State University of New York (SUNY); the University at Albany; the Atmospheric Sciences Research Center (ASRC) at the University at Albany; and the Department of Atmospheric and Environmental Sciences (DAES) at the University at Albany.

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
