# Peer review of "Quantitative Comparison of Power Production and Power Quality Onshore and Offshore: A Case Study from the Eastern U.S."

_Wind Energy Science, 2023_

## Referee Comment (RC1)

**Review of "Onshore and Offshore Wind Resources and Operating Conditions in the Eastern U.S." by Rebecca Foody, Jacob Coburn, Jeanie A. Aird, Rebecca J. Barthelmie and Sara C. Pryor**

This study describes the statistical analysis of a comprehensive lidar data set with focus on onshore and offshore wind speeds and power production in the U.S.

The data set described in this manuscript is interesting and of high relevance for wind energy research. The thematic itself is within the scope of the journal. However, a clear structure is missing, I could not identify clear objectives and/or hypotheses to be addressed and there is a lack of clear interpretations, discussions, and conclusions. The reader often needs either to accept statements without a clear proof or need to interpret the results by its own. Therefore, I recommend a major revision of the manuscript.

**Specific comments**

**Introduction:**
Entire Introduction: To my opinion, the big picture is missing here. What do you expect to find from the analyses. What do you want to explain or proof, what is your scientific question, the overall goal of your study?

Page 2, line 51: The authors mention that there are few previous studies without giving any references. I would also recommend to shortly summarize what has been done and found in those few studies.

Page 2, Point 1: I would expect a discussion/statement/explanation in the results part about the differences in the regions and what we can learn from it (e.g., an evaluation as to whether a region is more suitable as the others, beyond onshore-offshore differences)

Page 2, line 79: I could not find a guidance regarding optimal special scale. I don't feel guided with only a short statement that correlation is lower than 0.4 for distances >350 km. I recommend using either another formulation or provide a more detailed and profound discussion in the results part.

Page 3, paragraph 1: There is quite a harsh transition from the previous paragraph and topics to this one. The topic is completely new, and I miss a kind of introduction to why this is important in your study and what you want to show/discuss with results to this. What is your goal, what do you want to compare and what are possible consequences for your major hypothesis? In this paragraph, you mention something from structural loading and wakes but what does it mean for whatever you want to show and where is the discussion about it in the results/conclusion part?

Page 3, paragraph 2: Wouldn't it be better to paste a goal at the beginning of a paragraph? Otherwise, there is again a harsh transition from one topic to another from which the reader initially has no idea what the reason is, where to focus on.

**Data Sources:**
Page 4: The best year has a specific time frame for all positions, how about the analyses which are not based on the best years? Is the time span for all positions from January 2019 to December 2022 or are there some variabilities? If yes, how large are these and how would you expect them to influence your results?

**Methods:**
Page 7, chapter 3.3: Which question shall be answered by an analysis of the wind profile and why are you using shear and LLJ? In what sense are LLJ of relevance to wind energy applications?

Page 7, lines 223-225: Is this a commonly used method? Do you have references for this method which show that this can be done for wind energy or similar purposes? I wonder how suitable this method is considering the lower spatial and temporal resolution of ERA-5 compared to measurements, the

difference in time spans (comparing results from 44 years (ERA-5) to max 4 years (data) and the fact that ERA-5 has uncertainties on its own and additional uncertainties by the conversion from 10&100 m to 150 m. I would also suggest providing a kind of uncertainty or at least a discussion about this issue.

**Results:**
Page 8, lines 258-259: What does this mean? Why are low summer values a hint for a negative bias? Summer values are often lower, winter values often higher, so there are seasonal deviations from annual (and also long-term) mean values.

Page 8, lines 258-259: Definitely missing here is a detailed description and justification (preferably with reference) of how a long-term time series with relatively coarse resolution can lead to a meaningful error estimate of a point measurement, even more so when different time periods are used for this purpose. Why isn't it more likely here, that differences come from the interannual variability? What makes you believe that the data availability is responsible for these differences, in particular when you consider annual averages, and if so, wouldn't another way of calculating annual averages be the solution to avoid or at least minimize the influence? How do you calculate them that they have such a strong influence?

Page 11: Concerning the differences in Weibull scale parameters and AEP between best year and all data: What is the conclusion of this finding? Interannual variability? Data under-/overrepresentation? Any proofs for the one or the other?

Page 12, lines 290-293: A description/interpretation of figure 5 would be great. In general, the reader is a bit left alone with the interpretation of the figures. Either one understands it immediately on its own or not. Some help would be nice for all who didn't create the figures.

Page 13, lines 304-305: Could you explain this a bit more, please. There is an image, you could lead the reader through the image, just a bit, and let someone not being such deep into statistics see the same like you. Furthermore, do you have an explanation/expectation why the e-folding times at sea are larger than on land and why there is a slight difference in the onshore stations? E.g., any physical reasons for that?

Page 13, lines 308-309: Why does a large e-folding indicate the potential for more accurate power prediction? Are there any proofs? Did someone find this (citation?), or did you do any calculations?

Page 15: What is the conclusion from the analysis of shear conditions and LLJ?

Page 15/16, chapter demand matching: How did you calculate the normalized demand and site WPP and how do you relate it to the demand? What does it exactly mean, which conclusions can you draw from the findings? I guess, the couple of positions equipped with one turbine per position will not be able to cover the energy demand, but from the image it looks a bit like this. Means: further explanations are needed here to guide the reader into the right direction. And what does the comparison with ERA-5 reveal?

**Concluding remarks:**
At best, I see a summary here but no conclusion and no answer to a concrete scientific question or proof of a hypothesis. The reader is more or less left alone with the interpretation of this statistical analyses.

Page 16, lines 386-387: You compare the data to find out what? For what is it helpful, what does it aim for?

Page 17: 402-403: This is only a guess, I didn't see neither a proof nor a clear or understandable assessment of this (as has been stated in the introduction).

Page 17, line 410-411: To what extent does this follow from the Spearman correlation coefficients drop?

Page 17, line 414-420: The LCoE and its calculation was not mentioned in the results.

**Figures:**
Figure 3: For the comparison between onshore and offshore wind speeds I would suggest to create the same ranges for the y-axes. Also, the onshore figure is a bit crowded, maybe it would be a bit clearer to put the data availability into an own subfigure.

Figure 5: The figure is quite crowded, very small and differences are hard to interpret. I would also love to see a much better description in the text.

Figure 6: The lag time is in 10 Minute intervals, which needs an ad hoc recalculation to hours while reading the text, which in turn states the e-folding times in hours. I would recommend adapting either the text or, even better, the figures x-axis in a way that both becomes consistent.

Figure 8: Again very crowded, again, the image is not intuitively understandable without a more detailed description in the text.

---

## Author Comment (AC1)

Response to review

The reviewer's comments are given below in black font and our responses are in green font. A full tracked changes version of the manuscript is at the end of this document. Note also in responding to the reviewers comments (details below) we have also added 7 references.

Reviewer #1:

Review of "Onshore and Offshore Wind Resources and Operating Conditions in the Eastern U.S." by Rebecca Foody, Jacob Coburn, Jeanie A. Aird, Rebecca J. Barthelmie and Sara C. Pryor
This study describes the statistical analysis of a comprehensive lidar data set with focus on onshore and offshore wind speeds and power production in the U.S.
The data set described in this manuscript is interesting and of high relevance for wind energy research. The thematic itself is within the scope of the journal. However, a clear structure is missing, I could not identify clear objectives and/or hypotheses to be addressed and there is a lack of clear interpretations, discussions, and conclusions. The reader often needs either to accept statements without a clear proof or need to interpret the results by its own. Therefore, I recommend a major revision of the manuscript.
Response: We regret that the reviewer did not find our structure clear. We felt that by writing in the paragraph starting on line 51 'We evaluate four aspects of the wind power generation potential on- and offshore:' and then listing them that we were setting out the objectives clearly. But in light of your concerns, we have completely restructured the introduction.
We have also changed the title to help the reader immediately know the purpose of the manuscript the new title is **Quantitative Comparison of Power Production and Power Quality Onshore and Offshore: A Case Study from the Eastern U.S.**

Specific comments
Introduction:
Entire Introduction: To my opinion, the big picture is missing here. What do you expect to find from the analyses. What do you want to explain or proof, what is your scientific question, the overall goal of your study?
We have re-written the introduction to read (quoting from the revised manuscript):

[revised manuscript text omitted]

We feel this is a very clear statement of objectives, justification of the objectives and hope the reviewer concurs.

Page 2, line 51: The authors mention that there are few previous studies without giving any references. I would also recommend to shortly summarize what has been done and found in those few studies.

We actually couldn't find any comparable studies but did not want to preclude the possibility that such studies exist, but since we could not find a comparable studies we have dropped this statement.

Page 2, Point 1: I would expect a discussion/statement/explanation in the results part about the differences in the regions and what we can learn from it (e.g., an evaluation as to whether a region is more suitable as the others, beyond onshore-offshore differences)

We think the reviewer is referring to 'First, wind speeds tend to be higher and more consistent offshore due to both the lower surface roughness and lack of obstacles and topographic features that extract momentum and reduce both the wind speed and wind resource (Pryor and Barthelmie, 2002). Accordingly, Capacity Factors (CF), which are the ratio of actual power generation divided by the theoretical maximum power generation, are typically higher offshore. Data from operating wind farms in Denmark indicate CF from four offshore wind farms with installed capacity (IC) of 160 to 400 MW of 41-53% while CF from smaller onshore wind farms (IC: 16-70 MW) have CF of 28-41% (Enevoldsen and Jacobson, 2021). Within the U.S., the mean CF for onshore wind farms built between 2014 and 2019 is approximately 41% (Wiser et al., 2021). Simulations using numerical models for offshore wind energy lease areas along the U.S. east coast indicate CF above 46% largely as a result of the higher wind speeds offshore (Pryor et al., 2021;Barthelmie et al., 2023). '

It certainly is true that some regions of the world exhibit higher wind resources but our focus here is listing the factors that are responsible for 'Enhanced deployment of wind turbines offshore offers great promise in terms of enhanced renewable energy penetration into the electricity generation portfolio for three primary reasons'… So, respectfully, we are unconvinced that adding a discussion of the relative wind resource in different regions of the world would be useful here.

Page 2, line 79: I could not find a guidance regarding optimal special scale. I don't feel guided with only a short statement that correlation is lower than 0.4 for distances >350 km. I recommend using either another formulation or provide a more detailed and profound discussion in the results part.

Our apologies – the concept of the spatial decay of correlation and the e-folding distance may, indeed, not be widely understood. We have elaborated on this matter a little in the Methods. We have modified this sentence to read; 'Temporal autocorrelation coefficients of the power production time series are used to derive e-folding time scales (i.e. the time delay at which the correlation coefficient drops to $e^{-1}$,

i.e. to 0.37) which is used to represent the time scale at which the system 'loses' the memory of the initial state (Wilks, 2011).'
(i.e. to add 0.37)
And then have added this sentence:
'The e-folding concept can also be applied in this context, to quantify the distance at the power production from two sites is no longer significantly correlated.'
So the bottom line is the separation distance at which the correlation coefficient for the power production time series from two sites two drops below about 0.37 (rounding to 0.4) is the distance at which the sites are no longer significantly correlated – or if you prefer if I wish to achieve a more stable electricity production through time I should place wind farms at sufficient separation that their individual generation is not significantly correlated. For this wind climate that distance is about 350 km. We hope our changes to the text help to clarify that point.

Page 3, paragraph 1: There is quite a harsh transition from the previous paragraph and topics to this one. The topic is completely new, and I miss a kind of introduction to why this is important in your study and what you want to show/discuss with results to this. What is your goal, what do you want to compare and what are possible consequences for your major hypothesis? In this paragraph, you mention something from structural loading and wakes but what does it mean for whatever you want to show and where is the discussion about it in the results/conclusion part?

We regret you found this to be a harsh transition. Hopefully with the re-structure introduction you find the flow is smoother.

Page 3, paragraph 2: Wouldn't it be better to paste a goal at the beginning of a paragraph? Otherwise, there is again a harsh transition from one topic to another from which the reader initially has no idea what the reason is, where to focus on.

Again, we regret you found this to be a harsh transition. Hopefully with the re-structure introduction the flow is smoother. We do have an objective at the start of each bullet point.

Data Sources:

Page 4: The best year has a specific time frame for all positions, how about the analyses which are not based on the best years? Is the time span for all positions from January 2019 to December 2022 or are there some variabilities? If yes, how large are these and how would you expect them to influence your results?

There is always a compromise to be made – use all the data that you have in order to increase the sample size for statistical testing versus use the data period that best represents the seasonal cycle. This is indeed a challenge. So, for statistical testing where sample size aids confidence we use all records (e.g. spatial autocorrelation) but we also present to the reader information regarding how estimated AEP varies as a function of the data sample.

Methods:

Page 7, chapter 3.3: Which question shall be answered by an analysis of the wind profile and why are you using shear and LLJ? In what sense are LLJ of relevance to wind energy applications?

As we wrote:

'The International Electrotechnical Commission (IEC) 61400-1 standard states the expected value of $\alpha$ over land is 0.2 and is typically in the range of 0.05 to 0.25 and uses a value of 0.2 in the normal wind profile model (IEC, 2019). The occurrence of $\alpha$ beyond this range implies shear across the rotor plane differs from this design expectation and hence may indicate higher mechanical loading.

Thus on the most fundamental level we are seeking to report how frequently the shear is outside this expectation. The LLJ analysis is really to examine if one source of anomalous wind shear profiles (i.e. the frequency of LLJ) is higher/lower on and offshore.' We hope the modified introduction and the addition of an additional reference.

We have also added some text to the results section.: 'The implication is that large wind turbines deployed in these locations may experience a relatively high frequency of large unbalanced rotor loads and reduced component lifetimes unless such loads can be appropriately compensated (Hur et al., 2017).'

Page 7, lines 223-225: Is this a commonly used method? Do you have references for this method which show that this can be done for wind energy or similar purposes? I wonder how suitable this method is considering the lower spatial and temporal resolution of ERA-5 compared to measurements, the difference in time spans (comparing results from 44 years (ERA-5) to max 4 years (data) and the fact that ERA-5 has uncertainties on its own and additional uncertainties by the conversion from 10&100 m to 150 m. I would also suggest providing a kind of uncertainty or at least a discussion about this issue.

We believe the reviewer is referring to; 'Hourly values from the 40-year $U150_{ERA5}$ and $P150_{ERA5}$ record are randomly resampled 1000 times with replacement using the number of hours from each month that the LiDAR data are available (Figure 3). For each of these 1000 bootstrapped samples the annual mean wind speed and AEP is calculated to provide an estimate of uncertainty due to the short time series from the LiDARs.' Yes, bootstrap resampling is very frequently used to quantify confidence intervals around a metric (see for example the textbook by Dan Wilks or this text by Mudelsee; Climate Time Series Analysis: Classical Statistical and Bootstrap Methods (Atmospheric and Oceanographic Sciences Library, 51)).  We have added a citation of the Wilks reference to the methods.

Perhaps the reviewer is speaking to our specific application. The mean power law coefficient 0.21 which for a height interval of 100 to 150 is equal to a correction of 0.06 (or if you prefer a multiplier of 1.09 on the 100-m wind speed) so it's a small correction. We have now clarified this in the text. In terms of bootstrap resampling to derive uncertainties on wind speeds per se, it is a generalizable statistical method that can be applied to any geophysical property (see the book by Mudelsee)

We have clarified what the purpose of this analysis is by adding this statement; 'This analysis explicitly acknowledges the presence of low-frequency variability (seasonal to multi-decadal) in mid-latitude wind speeds and wind resources (Pryor et al., 2020a) and is designed to quantify the uncertainty on mean wind speeds and power production computed from the relatively short LiDAR data time series. '

Results:

Page 8, lines 258-259: What does this mean? Why are low summer values a hint for a negative bias? Summer values are often lower, winter values often higher, so there are seasonal deviations from annual (and also long-term) mean values.

We regret this statement wasn't clearer. We have rewritten it to read:

'Bootstrapping of ERA5 data indicates the mean annual wind speed computed from the LiDAR time series at the NYSM sites is likely underestimated by ~ 1.5-4.5% while AEP is underestimated by ~3-10% due to the high data availability in summer.'

We hope this clarifies.

Page 8, lines 258-259: Definitely missing here is a detailed description and justification (preferably with reference) of how a long-term time series with relatively coarse resolution can lead to a meaningful error estimate of a point measurement, even more so when different time periods are used for this purpose. Why isn't it more likely here, that differences come from the interannual variability? What makes you believe that the data availability is responsible for these differences, in particular when you consider annual averages, and if so, wouldn't another way of calculating annual averages be the solution to avoid or at least minimize the influence? How do you calculate them that they have such a strong influence?

We regret any confusion – precisely we are examining inter-annual variability!. We hope we have removed any confusion by adding this statement in methods: 'Although the LiDAR data sets that we analyze here are – to our knowledge – unique in terms of the duration and number of sites considered, we also contextualize the results and inferences drawn from these multi-year, but relatively short

duration, observations using the > 40 year duration ERA5 reanalysis product (Hersbach et al., 2020). This analysis explicitly acknowledges the presence of low-frequency variability (seasonal to multi-decadal) in mid-latitude wind speeds and wind resources (Pryor et al., 2020a) and is designed to quantify the uncertainty on mean wind speeds and power production computed from the relatively short LiDAR data time series. '

A minor note: If the cause is inter-annual variability due to differences in cyclone frequency/intensity that will be manifest in approximately equal magnitude in 'point' and spatially averaged values (unless the site is in complex terrain where directional channeling may be a factor).

Page 11: Concerning the differences in Weibull scale parameters and AEP between best year and all data: What is the conclusion of this finding? Interannual variability? Data under-/overrepresentation? Any proofs for the one or the other?

We did this analysis because it is important to acknowledge sources of uncertainty including incomplete time series. We note that in doing these analyses we also demonstrate that the on-shore off-shore differences are robust to data sampling issues. Accordingly, to avoid any confusion we have added the statement; 'It is important to note that the differences in energy density computed from the on-shore and off-shore LiDAR data sets are robust to these sampling issues.'

Page 12, lines 290-293: A description/interpretation of figure 5 would be great. In general, the reader is a bit left alone with the interpretation of the figures. Either one understands it immediately on its own or not. Some help would be nice for all who didn't create the figures.

We regret any confusion regarding interpretation of this figure. The definitions of ramps are given in equation (3) and accompanying text in Methods. We have expanded that description this a little to read: 'The probability of wind speed and power production ramp events are computed from the NYSERDA and NYSM LiDARs and in the case of wind speeds are normalized as follows:

$$\frac{\delta u(t)}{\sigma_{\delta u}} = \frac{u(t + \tau) - u(t)}{\sigma_{\delta u}} \tag{3}$$

where $u(t)$ is the wind speed at time $t$, $\delta u(t)$ is the wind speed increment from the prior time step, $\tau$ is the chosen time increment, and $\sigma_{\delta u}$ is the standard deviation of the wind speed increments (DeMarco and Basu, 2018). $\frac{\delta u(t)}{\sigma_{\delta u}} = 2$ indicates an increase in wind speed between two consecutive measurements (here $\tau$ = 10 minutes) of a magnitude that is equal to two standard deviations of wind speed changes computed from the entire time series, and thus lies in highest 2.5% of values. Conversely, $\frac{\delta u(t)}{\sigma_{\delta u}} = -2$, has a similarly low probability but is associated with a large magnitude decline in wind speed between two consecutive measurements.'

We have also expanded the paragraph that links to Figure 5 to read:

'The second component of power quality is the intermittency in terms of the probability and magnitude of ramp events – that is rapid changes in wind speed and/or power production. Wind speed time series at 150 m height from the NYSM and NYSERDA LiDARs indicate clear similarities in terms of ramp event magnitude and frequency to those derived using data from the FINO1 platform in the North Sea, Cabauw onshore in western portion of the Netherlands, Høvsøre in coastal Jutland, Denmark, and NWTC in the foothills of the Colorado Rocky Mountains (DeMarco and Basu, 2018) (Figure 5). Data from the NYSERDA buoys indicate a low probability of wind speed ramps of all magnitudes relative to the NYSM LiDARs (Figure 5), and all LiDAR time series indicate a substantially higher probability of a ramp-up (increase) than a ramp-down (decrease) of a given magnitude in wind speeds. Wind speed ramps in hourly ERA5 data exhibit a narrower distribution owing to spatial and temporal averaging, illustrating the need for in-situ data for capturing high resolution wind variability (Figure 5). Consistent with the lower probability of large-magnitude rapid changes in wind speed offshore, data from the NYSERDA buoys (Hudson North E05 and Hudson South E06) indicate probabilities of a wind power ramp with > ±20% change in power

are considerably lower than those from any of the onshore locations (Figure 5). Thus, the chance of experiencing an increase or decrease in electrical power production of 20% from one 10-minute period to the next is substantially lower for wind turbines deployed offshore.  This indicates that wind turbines deployed offshore are likely to exhibit less intermittency in terms of electrical power production which is critical to efficient grid integration (Ayodele et al., 2012).'

We have also slightly modified the caption to Figure 5 to read:

**'Figure 5. Left: probabilities of wind speed ramp events computed from the 10-minute data from the NYSERDA LiDAR buoys and the NYSM sites computed using Equation (3), and reported for four locations at or near operating wind turbines: FINO 1 is (offshore) in the North Sea, Cabauw is in the western portion of the Netherlands, Høvsøre is in Jutland, Denmark, and NWTC is in the foothills of the Colorado Rocky Mountains (data digitized from: (DeMarco and Basu, 2018)). Wind ramps computed from the hourly ERA5 output are shown by the gray polygon. Right: probabilities of wind power production ramp events at the locations of the NYSERDA buoys and the NYSM sites computed by applying the power curve for the IEA 15 MW reference wind turbine to the LiDAR wind speeds. The probabilities of no-change (i.e., power ± 0%) are not shown to aid visibility.'**

Page 13, lines 304-305: Could you explain this a bit more, please. There is an image, you could lead the reader through the image, just a bit, and let someone not being such deep into statistics see the same like you. Furthermore, do you have an explanation/expectation why the e-folding times at sea are larger than on land and why there is a slight difference in the onshore stations? E.g., any physical reasons for that?

The reviewer is referring to the following: 'The third aspect of power quality is predictability. The autocorrelation in power production at different time lags for the NYSM LiDARs exhibit clear diurnal oscillations and shorter e-folding time scales.'

We hope that adding text in Introduction

'Predictability and persistence of wind speeds and power production (Haghi et al., 2013;Haslett and Raftery, 1989). Within liberalized electricity markets, wind farm owner/operators bid in advance (e.g. 24 hours in advance) and are charged penalties for any imbalance between the bid and actual production (Pinson et al., 2007). Hence, accurate forecasts of wind generation are important to reduce penalties and maximizing revenue (Barthelmie et al., 2008). Persistence models where the power production at some future time is model as a function of power production in the recent past is often used as a benchmark forecast against which more sophisticated short-term power production models are compared (Kariniotakis et al., 2004). Many statistical short-term forecast models are predicated in part on persistence (Zeng and Qiao, 2011) and thus are most skillful when the power production time series exhibits high temporal autocorrelation. We quantify the temporal autocorrelation of power production from each onshore and offshore site and compare the degree to which electrical power production from the onshore and offshore locations differ with respect to persistence and short-term predictability.'

And in the Methods to this sentence:

'Temporal autocorrelation coefficients of the power production time series are used to derive e-folding time scales (i.e. the time delay at which the correlation coefficient drops to $e^{-1}$, i.e. to 0.37) which is used to represent the time scale at which the system 'loses' the memory of the initial state (Wilks, 2011).'

And this where we discuss Figure 5:

'These relatively large e-folding times for the buoy locations indicate a longer atmospheric 'memory' at these sites, indicating the potential for more accurate short-term power prediction forecasts because each time step is strongly dependent on previous time step(s).'

Clarifies this matter.

Page 13, lines 308-309: Why does a large e-folding indicate the potential for more accurate power prediction? Are there any proofs? Did someone find this (citation?), or did you do any calculations?

Please see the answer directly above.

Page 15: What is the conclusion from the analysis of shear conditions and LLJ?

We have added these two sentences that we believe makes the inference more concrete:

'The implication is that large wind turbines deployed in these locations may experience a relatively high frequency of large unbalanced rotor loads and reduced component lifetimes unless such loads can be appropriately compensated (Hur et al., 2017).'

And

'It is important to acknowledge that comparisons of LLJ climates derived from LiDAR measurements and WRF modelling should be done cautiously and that LLJ detection from the LiDAR wind speed profiles is critically dependent on unbiased data availability. Nevertheless, this analysis suggests LLJ within the rotor plane, as a source of large unbalanced rotor loads and reduced blade lifetimes, are less frequent at these onshore locations. '

Page 15/16, chapter demand matching: How did you calculate the normalized demand and site WPP and how do you relate it to the demand? What does it exactly mean, which conclusions can you draw from the findings? I guess, the couple of positions equipped with one turbine per position will not be able to cover the energy demand, but from the image it looks a bit like this. Means: further explanations are needed here to guide the reader into the right direction. And what does the comparison with ERA-5 reveal?

We calculate the demand as follows (see Methods):

'Electrical demand (in MWh) for New York state are also presented and mean values are computed for each hour of the day and each month of the year based on hourly values for 2016-2022 as reported by the U.S. Energy Information Administration (EIA) hourly electric-grid monitor (https://www.eia.gov/electricity/gridmonitor/dashboard/electric_overview/US48/US48)'.

And we normalize it as noted in the caption to Figure 9: 'The data are normalized to a mean value of 1 and so that values of 0.9 or 1.1 in a given hour or month indicates WPP or demand that is 10% below or above the mean, respectively.'

We have expanded the paragraph that relates to Figure 9 to read as follows:

'Electricity demand in New York state tends to peak in the afternoon (~ 1700 eastern standard time, EST) and in summer (highest values in July), though a secondary maximum occurs in January (Figure 9). Wind power production calculated from the NYSM/NYSERDA LiDARs and ERA5 grid-cell data ($P150_{ERA5}$) show highest values at night (0100 to 0500 EST) and during winter to spring (December-April), with the lowest production during the day (1300 to 1600 EST) and during the summer (Figure 9). Wind power production estimated based on LiDAR data from the NYSERDA buoy locations exhibits markedly lower diurnal and seasonal variability than is estimated at the NYSM sites, varying by ± 10% around the mean versus ± 25% at NYSM. This results in a reduction in mean absolute error (MAE) between time series of normalized WPP from the offshore LiDAR and electricity demand on both diurnal and seasonal timescales. The MAE computed from the mean hourly offshore WPP and demand is 0.19 when computed over the 24 hours of the day (Figure 9a) and 0.13 when computed from the time series of monthly mean values (Figure 9b). Both are smaller than MAE computed from WPP from the onshore LiDARs and demand on these time scales which are 0.25 and 0.20, respectively. This implies there will be better matching to electricity demand for power production from wind turbines deployed offshore. '

We hope this clarifies.

Concluding remarks:

At best, I see a summary here but no conclusion and no answer to a concrete scientific question or proof of a hypothesis. The reader is more or less left alone with the interpretation of this statistical analyses.

We regret the reviewer found this section challenging to interpret.  We have supplemented materials in the conclusions to help the reader. The conclusions now read:

[revised manuscript text omitted]

Page 16, lines 386-387: You compare the data to find out what? For what is it helpful, what does it aim for?

We hope the rewrite of the introduction and conclusion helps – we are seeking to quantify the relative benefits of deploying wind turbines offshore.

Page 17: 402-403: This is only a guess, I didn't see neither a proof nor a clear or understandable assessment of this (as has been stated in the introduction).

No its not a guess the statement "higher temporal autocorrelation of wind power production offshore (Figure 6 and Table 2) may also aid the accuracy of short-term wind power forecasting" can be readily made because virtually all statistical short-term forecasting methods employ directly or indirectly methods that are predicated on red noise characteristics of the atmosphere. See additional text that explains that matter.

Page 17, line 410-411: To what extent does this follow from the Spearman correlation coefficients drop?

Exponential decay of atmospheric properties in time and space is well documented. We hope that explanatory text we have added about this matter helps.

Page 17, line 414-420: The LCoE and its calculation was not mentioned in the results.

We have moved some discussion of LCoE into section 4.1

Figures:

Figure 3: For the comparison between onshore and offshore wind speeds I would suggest to create the same ranges for the y-axes. Also, the onshore figure is a bit crowded, maybe it would be a bit clearer to put the data availability into an own subfigure.

We do not concur with the reviewer. We think there is importance to having the reader readily be able to note the data availability with the monthly mean wind speeds.

Figure 5: The figure is quite crowded, very small and differences are hard to interpret. I would also love to see a much better description in the text.

We have elaborated the text as requested.

Figure 6: The lag time is in 10 Minute intervals, which needs an ad hoc recalculation to hours while reading the text, which in turn states the e-folding times in hours. I would recommend adapting either the text or, even better, the figures x-axis in a way that both becomes consistent.

Text changes enacted as requested.

Figure 8: Again very crowded, again, the image is not intuitively understandable without a more detailed description in the text.

We regret the reviewer struggled to understand this figure. We have added this text to the caption:

**'A LLJ frequency of 5% calculated from the LiDAR deployed at QUEE for the calendar month of May indicates that LLJ were indicated in 5% of all 10-minute periods during this month. Data in panels (a) and (b) indicate that at that site in the month of May the associated LLJ mean core wind speed is 11 ms$^{-1}$ and the mean core height above ground is 330 m.'**

Reviewer #2:

General comments:

The paper presents data from a set of onshore and offshore lidars. The paper mostly presents statistics based on these data and the results of the analysis are as expected, as the wind blows more and more steady offshore compared to onshore. There is no new methods, concepts or ideas introduced in the manuscript, so I have rated the scientific significance as low. Nonetheless, the paper could be useful for somebody that is looking specifically for information about the wind climate in this region.

We regret the reviewer did not find the scientific significance to be higher. The lack of measurements at

wind turbine hub-heights and above has long been a source of concern for the wind energy community so we believe that the availability of network based LIDAR data will be of general interest. Further, we have found no previous research that used these types of data sets to quantitatively compare; the resource, power quality, demand matching etc on – and offshore in the same climate zone. And yet these properties are of critical importance in charting the future of wind energy deployments.

My main comment on the analysis itself is about the low data recovery percentage of the lidar data. It is not demonstrated that there is no correlation of when data recovery is low and what the wind climate is. For example, one would expect that the lidars return 'not available' when a measurement cannot be obtained. Most of these data will be during low wind speed conditions when there is not enough aerosols to measure the wind. You have your long-term measurement time series from ERA5, so you could correct for this. Also in general I miss some discussion of the type of lidars you are using, because they are not the same offshore (zephyr) and onshore (windcube). What kind of filtering was done (precipitation? CNR?).

It is indeed regrettable that the NYSM network does not have higher data recovery. We asked this question of the operators of the NYSM but they are not able to provide further information beyond what is available via the readme documentation (available at: http://www.nysmesonet.org/networks/profiler#stid=prof_alba) which simply states: 'Sensor and/or system failures are not uncommon as the Profiler equipment are sensitive to a variety of environmental factors. Data gaps may be due to sensor failures; calibration errors; power failures; and/or communication failures. … Only manufacturer-developed QA/QC procedures are applied to the data and there might still be some undetected errors.'

More information is available regarding the NYSERDA buoy deployments – e.g. via technical reports available for download from: https://oswbuoysny.resourcepanorama.dnv.com/download/f67d14ad-07ab-4652-16d2-08d71f257da1 We have added a note to that effect to the Data availability statement: 'Reports documenting LiDAR performance verification are also available for download from: https://oswbuoysny.resourcepanorama.dnv.com/download/f67d14ad-07ab-4652-16d2-08d71f257da1.'

Regarding the comment 'It is not demonstrated that there is no correlation of when data recovery is low and what the wind climate is.' We think there are two concepts of importance here:

1) We do document at the monthly scale data availability versus wind speed.
2) There is no documented evidence from NYSM that the LiDAR scans failed to report wind speeds due to low aerosol concentrations but given they state 'manufacturers QA/QC procedures were employed it is highly likely a CNR screen was employed'. That having been said air quality measurements in New York state DO NOT imply a high frequency of sufficiently low aerosol concentrations to render these LiDARs likely to be unable to operate (see for example; Squizzato, S., Masiol, M., Rich, D. Q., & Hopke, P. K. (2018). A long-term source apportionment of PM2. 5 in New York State during 2005–2016. *Atmospheric Environment*, *192*, 35-47.)

We have added this text to section 4.1; 'Documentation associated with the data set notes the causes as; 'calibration errors; power failures; and/or communication failures.' And further notes 'Only manufacturer-developed QA/QC procedures are applied to the data and there might still be some undetected errors.' (readme accessible from http://www.nysmesonet.org/networks/profiler#stid=prof_alba).'

Technical comments:

l13: factor -> parameter

Done.

l166: conventional usage would be capital gamma

changed as requested.

l211-215: since the lidar signal depends on aerosol concentration this method will likely miss many low-level jets as the lidar will simply not return a signal above the jet. Would be good to discuss this.

We have added this cautionary text:

'It is important to acknowledge that comparisons of LLJ climates derived from LiDAR measurements and WRF modelling should be done cautiously and that LLJ detection from the LiDAR wind speed profiles is critically dependent on unbiased data availability. Nevertheless, this analysis suggests LLJ within the rotor plane, as a source of large unbalanced rotor loads and reduced blade lifetimes, are less frequent at these onshore locations.'

In addition: I am not quite sure how to interpret the comment about the comparibility with the 500 m height: did you use data up to 500 m? It would be good to show what the recovery percentage is at this height, related to the remark above.

We have calculated the data recovery rates at each LiDAR location at each height < 500 m and now report them, see text that's reads; 'Analyses of wind speed data from the NYSM LiDARs at all measurement heights from 100 to 500 m indicates that averaged across all stations the data availability as a function of height varies only by +/-2.5%.'

l232: move bracket from before Barthelmie to before 2023.

Done.

[revised manuscript text omitted]

---

## Referee Report (RR1)

**Paper Review Quantitative Comparison of Power Production and Power Quality Onshore and Offshore: A Case Study from the Eastern U.S**.

Author: Rebecca Foody

**General**

This paper presents an observational analysis of potential onshore and offshore wind power production using data from two buoy-based LiDAR systems (sponsored by the New York State Energy Research and Development Authority, or NYSERDA) located in the New York Bight, and from several sites in the New York State Mesonet profiler network. The authors also complement the short-term LiDAR observation data sets with extrapolated winds from the longer-term ERA5 reanalysis product. Key points of the analysis include that 1) the offshore waters in the New York Bight, as characterized by the NYSERDA buoys, provide a significantly greater wind resource, 2) the offshore wind resource is more persistent (less intermittent) as compared with land-based (and even coastal) observations, 3) there is a summer peak in the frequency of low level jets (LLJs) and higher rotor plane shear, 4) not surprisingly, given the mid-latitude east coast location, geographic diversity (here defined has > 350 km) would reduce the potential for large-scale "wind power droughts", and 5) offshore wind is a more favorable location for load matching given reduced diurnal range of hub height winds and the large coastal populations in the region. Overall, the paper provides a very useful and cogent comparative analysis of the onshore and offshore (potential) wind resource in New York and the adjacent coastal waters. With minor edits, I recommend the draft manuscript for publication.

**Specific comments**

Page 2, line 45: note that aesthetics—visual blight, commercial fisheries, social equity, and NIMBY (e.g., transmission cable land fall) are significant social barrier issues for offshore wind siting.

Page 3, line 84: in addition to the Aird et al. (2022) paper, McCabe and Freedman (2023; see https://journals.ametsoc.org/view/journals/wefo/38/4/WAF-D-22-0119.1.xml) also recently published on the frequency and physical characteristics of the sea breeze and associated LLJ in the New York Bight and coastal NY (also using the NYSERDA and NYSM LiDARs).

Page 5, 174-176: this sentence is confusing— the ERA5 hourly data "…represent approximately 15- to 20-minute average values…."

Page 7, line 226: essentially $y \equiv r$?

Page 8, line 241: see McCabe and Freedman (2023)

Page 8, line 247: typo or grammar ("…all ERA5 grid-cells in [sic] that contain NYSM….").

Page 8, lines 246 - 259: should be more discussion of the limitations of using ERA5 data (perhaps in section 2.3?)—especially given the use of extrapolating using the calculated shear exponent between 10 m and 100 m. The co-authors of this paper have used ERA5 data sets in previous analyses, and other papers have discussed the issue of ERA5 underestimating near-surface wind speeds and smoothing out potential LLJ profiles (e.g., Kalverla et al. 2020).

Page 8, line 265: the cost recovery factor, CRF is mentioned once and does not appear in Table 1 but is part of the calculation in equation (8). It is defined in Barthelmie et al. (2023).

Page 12, Figure 3: top left graphic is tough to read.

Page 14, Figure 5: tough to clearly see Hudson North and Hudson South on right side figure.

Page 17, lines 410 - 420: should reference McCabe and Freedman (2023) on climatology of the LLJ. Compare and contrast their methods for identifying LLJs.

Page 18, Figure 9: should make this figure larger so can be read more easily.

---

## Author Response (AR2)

Response to review

The reviewer's comments are given below in black font and our responses are in green font. A full tracked changes version of the manuscript is at the end of this document.

Paper Review Quantitative Comparison of Power Production and Power Quality Onshore and Offshore: A Case Study from the Eastern U.S.
Author: Rebecca Foody
General
This paper presents an observational analysis of potential onshore and offshore wind power production using data from two buoy-based LiDAR systems (sponsored by the New York State Energy Research and Development Authority, or NYSERDA) located in the New York Bight, and from several sites in the New York State Mesonet profiler network. The authors also complement the short-term LiDAR observation data sets with extrapolated winds from the longer-term ERA5 reanalysis product. Key points of the analysis include that 1) the offshore waters in the New York Bight, as characterized by the NYSERDA buoys, provide a significantly greater wind resource, 2) the offshore wind resource is more persistent (less intermittent) as compared with land-based (and even coastal) observations, 3) there is a summer peak in the frequency of low level jets (LLJs) and higher rotor plane shear, 4) not surprisingly, given the mid-latitude east coast location, geographic diversity (here defined has > 350 km) would reduce the potential for large-scale "wind power droughts", and 5) offshore wind is a more favorable location for load matching given reduced diurnal range of hub height winds and the large coastal populations in the region. Overall, the paper provides a very useful and cogent comparative analysis of the onshore and offshore (potential) wind resource in New York and the adjacent coastal waters. With minor edits, I recommend the draft manuscript for publication.

Specific comments
Page 2, line 45: note that aesthetics—visual blight, commercial fisheries, social equity, and NIMBY (e.g., transmission cable land fall) are significant social barrier issues for offshore wind siting.
Response: For visual blight and noise concerns are generally lower than on land. Nevertheless, we have slightly modified this statement to include co-use etc.  Sentence modified to read:
> Second, there are generally fewer social barriers than exist on land (e.g., competition for land, noise concerns, visual blight, etc.) (Diógenes et al., 2020), although there are considerations regarding co-use (e.g. for commercial fishing and marine navigation) (Stone et al., 2017;Kirkegaard et al., 2023).

Page 3, line 84: in addition to the Aird et al. (2022) paper, McCabe and Freedman (2023; see https://journals.ametsoc.org/view/journals/wefo/38/4/WAF-D-22-0119.1.xml) also recently published on the frequency and physical characteristics of the sea breeze and associated LLJ in the New York Bight and coastal NY (also using the NYSERDA and NYSM LiDARs).
Response: Thanks for pointing it out. We have added the reference!

Page 5, 174-176: this sentence is confusing— the ERA5 hourly data "…represent approximately 15- to 20-minute average values…."
Response: rewritten to:
> Herein we analyze once-hourly estimates of the u- and v- wind components at a height of 100 m. We use ERA5 output for the period of record with highest quality data assimilated into the reanalysis system: 1979-2022. This time period also includes the observational period of the LiDARs.

Page 7, line 226: essentially $y \equiv r$?
Response: yes (we have added a parenthetical note to this effect)

Page 8, line 241: see McCabe and Freedman (2023)
We have included this text:

> Alternative metrics to detect LLJs have been proposed, including use of normalized wind increments by the height interval (i.e. a shear definition) (Hallgren et al., 2023;McCabe and Freedman, 2023).

Page 8, lines 246 - 259: should be more discussion of the limitations of using ERA5 data
Response: We are using ERA5 to examine low-frequency climatological variability but have added a cautionary note earlier (in section 2.3) we have added:

> However, past research has also indicated substantial spatio-temporal variability in the fidelity of ERA5 wind speed products of relevance to wind energy contexts (Pryor et al., 2020b;Kalverla et al., 2020;Meyer and Gottschall, 2022;Knoop et al., 2020). Here we are using ERA5 output to (i) examine climatological variability and thus contextualize the short observational records, (ii) provide context for the spatial decay of association manifest in the remote sensing observations and (iii) quantify the bias in annual mean wind speeds due to seasonal bias in LiDAR data availability.

(perhaps in section 2.3?)—especially given the use of extrapolating using the calculated shear exponent between 10 m and 100 m. The co-authors of this paper have used ERA5 data sets in previous analyses, and other papers have discussed the issue of ERA5 underestimating near-surface wind speeds and smoothing out potential LLJ profiles (e.g., Kalverla et al. 2020).
Response- yes see above we have added a cautionary note saying ERA5 exhibit variability in terms of fidelity.

Page 8, line 265: the cost recovery factor, CRF is mentioned once and does not appear in Table 1 but is part of the calculation in equation (8). It is defined in Barthelmie et al. (2023).
Response- sorry for the typo that made it seem that this parameter was not specified (it is in the Table).

Page 12, Figure 3: top left graphic is tough to read.
Response-We have revised to figure to make it clearer.

Page 14, Figure 5: tough to clearly see Hudson North and Hudson South on right side figure.
Response-We have also revised this figure to make it clearer.

Page 17, lines 410 - 420: should reference McCabe and Freedman (2023) on climatology of the LLJ. Compare and contrast their methods for identifying LLJs.
Response-We have included this text where we discuss how we define a LLJ:

> However, alternative metrics to detect LLJs have been proposed, including use of normalized wind increments by the height interval (i.e. a shear definition) (Hallgren et al., 2023;McCabe and Freedman, 2023).

Hallgren et al. (2023) does a great job of this comparison, so we will not duplicate that material here.

Page 18, Figure 9: should make this figure larger so can be read more easily.
Response-Done!

---

## Author Response (AR3)

Response to comments from EiC. 18 December 2023.

**Comments to the author**:

Dear authors,

Thanks for improving your manuscript which is also ready for publication. I would like you to expand on the impact of poor lidar availability on your results. This could be done by referencing to Carrier-to-Noise-Threshold Filtering on Off-Shore Wind Lidar Measurements Gryning, SE and Floors, R Feb 1 2019 | 19 (3) Sensors.

I'm not sure you address this issue as much as it deserves.

Sincerely, Jakob Mann

Our response:

We wish there was a clearer statement regarding CNR thresholds used for the two data sets. It is, as you suggest, important. At your request we have:

Added this text to section 2 (Data sources).

> A critical determinant of LiDAR-derived wind speed and direction climates is the carrier-to-noise ratio (CNR) used in quality control procedures. CNR is the ratio of the received carrier strength to the intensity of the received noise. Larger values imply higher measurement accuracy but there is ambiguity in terms of the optimal CNR threshold to ensure high wind climate fidelity. Early research with coherent continuous-wave wind LiDAR proposed use of a –22 dB CNR threshold to screen out periods with unacceptably high wind speed uncertainty (Frehlich, 1996), and this threshold has subsequently been widely adopted (Bischoff et al., 2017). Detailed analyses of measurements to 600 m height with Leosphere WLS70 pulsed Doppler LiDAR relative to sonic anemometers, found use of a –22 dB CNR threshold caused a 7 to 12 % overestimation in the long-term mean wind speed, with the higher discrepancy over coastal and marine sites (Gryning et al., 2016). A more recent study, using data from the Leosphere WLS70 deployed on the FINO platform in the North Sea, found a high sensitivity of the wind rose and mean wind speed to use of thresholds lower than –29 dB (Gryning and Floors, 2019). That analysis further found that for heights of 100 to 200 m, application of a –22 dB CNR threshold caused a 12% overestimation of mean wind speed, which decreased to 9% when a CNR threshold value of –35 dB was applied (Gryning and Floors, 2019). Optimal CNR thresholds may vary with site conditions and instrument. Use of different thresholds will influence only data quality but also data availability.

Added this text to the conclusions:

> The differences in wind climates, energy density and estimated power production from the offshore and onshore LiDAR are of sufficient magnitude that they likely exceed any discrepancy due to application of different CNR thresholds in data screening proceedures for the two LiDAR networks.

As implied by the above we now cite 3 additional references (Bischoff et al. was previously cited):

Bischoff, O., Wurth, I., Gottschall, J., Gribben, B., Hughes, J., Stein, D., and Verhoef, H.: Floating Lidar Systems, IEA Expert Group Report on Recommended Practices, IEA Wind TCP RP 18 from Task 32. Available for download from: https://iea-wind.org/portfolio-item/recommended-practice-18/, 89, 2017.

Frehlich, R.: Simulation of coherent Doppler lidar performance in the weak-signal regime, Journal of Atmospheric and Oceanic Technology, 13, 646-658, 1996.

Gryning, S.-E., Floors, R., Peña, A., Batchvarova, E., and Brümmer, B.: Weibull wind-speed distribution parameters derived from a combination of wind-lidar and tall-mast measurements over land, coastal and marine sites, Boundary-Layer Meteorology, 159, 329-348, 2016.

Gryning, S.-E., and Floors, R.: Carrier-to-noise-threshold filtering on off-shore wind lidar measurements, Sensors, 19, 592, doi: 510.3390/s19030592, 2019.